# House of Cards: Massive Weights in LLMs

## Abstract

Massive activations, which manifest in specific feature dimensions of hidden states, introduce a significant bias in large language models (LLMs), leading to an overemphasis on the corresponding token. In this paper, we identify that massive activations originate not from the hidden state but from the intermediate state of a feed-forward network module in an early layer. Expanding on the previous observation that massive activations occur only in specific feature dimensions, we dive deep into the weights that cause massive activations. Specifically, we define *top-$k$ massive weights* as the weights that contribute to the dimensions with the top-$k$ magnitudes in the intermediate state. When these massive weights are set to zero, the functionality of LLMs is entirely disrupted. However, when all weights except for massive weights are set to zero, it results in a relatively minor performance drop, even though a much larger number of weights are set to zero. This implies that during the pre-training process, learning is dominantly focused on massive weights. Building on this observation, we propose a simple plug-and-play method called `MacDrop` (**ma**ssive weights **c**urriculum **drop**out), to rely less on massive weights during parameter-efficient fine-tuning. This method applies dropout to the pre-trained massive weights, starting with a high dropout probability and gradually decreasing it as fine-tuning progresses. Through various experiments, including zero-shot downstream tasks, long-context tasks, and ablation studies, we demonstrate that `MacDrop` generally improves performance and strengthens robustness.

## 1. Introduction

Large language models (LLMs), such as GPT (Achiam et al., 2023) and Llama (Touvron et al., 2023; Dubey et al., 2024), have achieved remarkable success across diverse natural language tasks (Roziere et al., 2023; Mitra et al., 2024; Labrak et al., 2024; Wu et al., 2023). Their success is largely attributed to the pre-training phase, during which they are trained on extensive high-quality corpora datasets to predict the next token (Longpre et al., 2024; Zhao et al., 2024; Shen et al., 2023). However, despite the impressive achievements of LLMs, a crucial gap remains in our understanding of the underlying mechanisms that drive their remarkable performance.

Recently, Xiao et al. (2024) uncovered an intriguing phenomenon in LLMs, referred to as *attention sinks*: an unexpectedly large portion of attention is directed toward the initial tokens, regardless of their semantic context, after a small number of early layers. They demonstrated that under a restricted budget, focusing attention solely on recent window leads to poor performance, and that performance is recovered when initial tokens are included. Based on this observation, they proposed StreamingLLM, which retains the key-value caches of the initial sink tokens and the recent tokens for streaming use of LLMs. Yu et al. (2024b) further investigated the attention sinks phenomenon, finding that attention sinks can appear both in the initial tokens and in later tokens with less semantic importance (e.g., '.' and '\n'). They showed that when sink tokens appear later in a sequence, sink tokens can potentially result in performance degradation. Inspired by this observation, they proposed a head-wise attention calibration technique without requiring additional training. Concurrently, Sun et al. (2024a) discovered the existence of *massive activations* in the hidden states of LLMs, with magnitudes substantially larger than the others. Massive activations are jointly identified based on their sequence and feature dimensions within the hidden states. Specifically, massive activations occur at the initial tokens and weak semantic tokens according to the model, and are consistently present in only a few fixed feature dimensions. Moreover, they connected massive activations with attention sinks, suggesting that massive activations inject implicit bias into the self-attention mechanism throughout the pre-training phase.

---

[1]Anonymous Institution, Anonymous City, Anonymous Region, Anonymous Country. Correspondence to: Anonymous Author <anon.email@domain.com>.

Preliminary work. Under review by the International Conference on Machine Learning (ICML). Do not distribute.

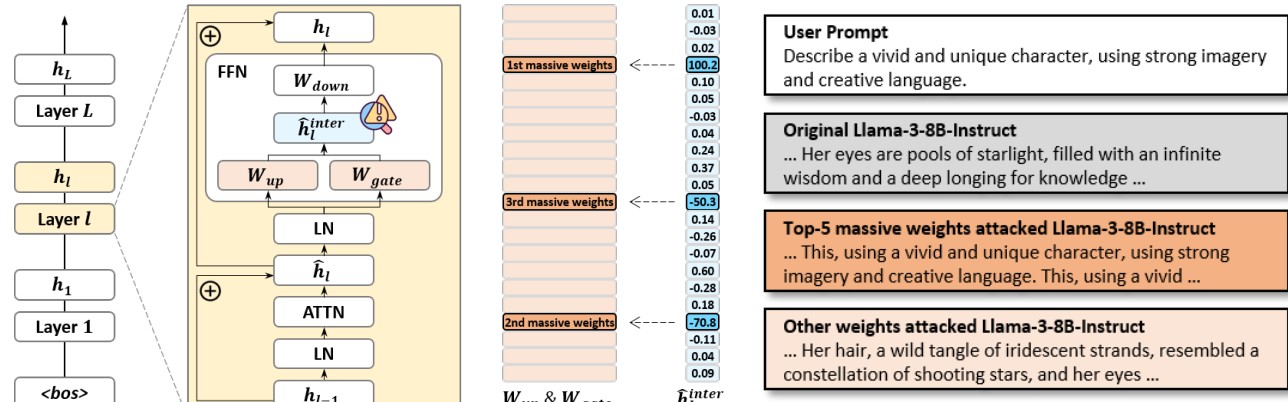

(a) Massive weights.

(b) Examples of generated responses.

Figure 1. (a) Massive weights are defined as the rows of $W_{gate}$ and $W_{up}$ in a specific layer $l$ using the *bos* token, which produce the top-$k$ magnitudes of the intermediate state $\hat{h}_l^{inter}$. Because massive weights are defined within a single layer $l$, the ratio of massive weights is significantly low compared to the overall number of parameters. For instance, in the case of Llama-3-8B, the proportion of the top-5 massive weights is 0.0005% of the model's total parameters. (b) When the top-5 massive weights are zeroed out, instruction-tuned LLMs completely lose their ability to generate text. On the other hand, when only the top-5 massive weights remain unchanged in $W_{gate}$ and $W_{up}$, instruction-tuned LLMs retain their generation capability.

In this paper, we first delve deeper into massive activations, providing two key observations. (1) The *bos* token placed at the starting position always has massive activations in the same feature dimensions and makes attention sinks. This observation enables us to focus only on the feature dimensions, instead of both the sequence and feature dimensions, when addressing massive activations. Namely, a simplified and consistent analysis of massive activations can be achieved by using only the *bos* token. (2) Massive activations originate in the intermediate state $\hat{h}_l^{inter}$ within an early layer $l$, before appearing in the hidden state $h_l$, as illustrated in Figure 1(a). Namely, massive activations triggered in $\hat{h}_l^{inter}$ are subsequently and continuously propagated through skip connections. This observation implies that the feed-forward network in layer $l$ plays a crucial role in LLMs.

Next, we shift our focus from activations to the weights, relying on the fact that massive activations consistently appear in the same feature dimensions. In detail, we define the *top-k massive weights* as the rows of $W_{up}$ and $W_{gate}$ in the feed-forward network at layer $l$ that produce the top-$k$ magnitudes of the intermediate state $\hat{h}_l^{inter}$, as illustrated in Figure 1(a). It is important to note that massive weights, defined within a single layer $l$, account for a substantially small fraction compared to the model's total parameters. This holds true even when compared to the entire $W_{up}$ and $W_{gate}$. Nevertheless, massive weights are crucial factors that can completely influence the performance of LLMs. Figure 1(b) presents the generated responses of three models to the given user prompt: original model, top-5 massive weights attacked model, and other weights attacked model. Here, other weights represent all weights in $W_{up}$ and $W_{gate}$ at layer $l$ that do not belong to the top-5 massive weights,

and an attack sets corresponding weights to zero. When the massive weights are attacked, the model becomes poor and repeats the user prompt. On the contrary, when other weights are attacked, the model does not entirely lose its generation capability, even though a much greater number of weights are set to zero in the same projection matrices. These observations imply that massive weights are dominantly learned during pre-training and highly related to the performance of LLMs.

Finally, we propose a straightforward plug-and-play method during parameter-efficient fine-tuning, named **ma**ssive weights **c**urriculum **drop**out (`MacDrop`). This method applies dropout to the pre-trained massive weights, rather than additional trainable weights, starting with a high dropout rate that is progressively reduced throughout the fine-tuning phase. The intuition behind `MacDrop` is that a high initial dropout rate encourages the model to lessen dependence on the massive weights predominantly learned during the pre-training phase. Then, reducing the dropout rate facilitates a more stable convergence, ensuring the pre-trained model is leveraged with neglectable damage by the end of fine-tuning. Through various ablation studies, we examine the effects of dropout scope and dropout probability scheduling. Finally, we demonstrate that `MacDrop` generally enhances model performance and robustness in zero-shot downstream tasks and long context tasks.

## 2. Massive Weights

In this section, we review the key observations on massive activations reported by Sun et al. (2024a) and extend the analysis by exploring various states using the *bos* token,

which was not covered thoroughly. Based on this expanded analysis, we formally define top-$k$ massive weights in a specific layer and investigate their importance through two opposite types of attacks.

## 2.1. Prerequisite: Massive Activations

Autoregressive Transformers (Vaswani et al., 2017) are structured with $L$ decoding layers. Each layer $l \in [1, L]$ includes an attention (ATTN) module and a feed-forward network (FFN) module. These modules are connected via residual connections (He et al., 2016), each following a layer normalization (LN) layer (Ba, 2016). The previous hidden state $h_{l-1}$ is fed into layer $l$ and processed to produce the subsequent hidden state $h_l$:

$$h_l = \hat{h}_l + \text{FFN}(\text{LN}(\hat{h}_l)),$$
$$\text{where } \hat{h}_l = h_{l-1} + \text{ATTN}(\text{LN}(h_{l-1})) \tag{1}$$

Sun et al. (2024a) primarily concentrated on the activations within hidden states, identifying that certain activations exhibit exceptionally large magnitudes, which they termed *massive activations*. Massive activations are observed at the starting position (i.e., input-agnostic) or at the delimiter tokens, depending on the model. Furthermore, these activations are confined to a small number of fixed feature dimensions, even within these tokens. These activations initially emerge after passing through several early layers and then decreases as they near the last layer.

Massive activations are strongly tied to the attention sinks phenomenon, as identified by Xiao et al. (2024), in which attention is abnormally concentrated on a small subset of tokens. In detail, a given query state tends to have positive cosine similarity with the key states of the tokens exhibiting massive activations, and negative cosine similarity with those of other tokens. Consequently, attention is heavily skewed toward the tokens associated with massive activations. Detailed related work is explained in Appendix H.

## 2.2. Further Analysis on Massive Acitvations

We primarily utilize the Llama-3-8B model (Dubey et al., 2024) and explicitly specify other models when necessary.

***bos* token placed at the starting position always has massive activations.** We begin by examining whether any specific condition consistently triggers massive activations. The existence of such a condition would greatly facilitate the analysis and algorithm development for handling massive activations. Following Sun et al. (2024a), massive activations are observed when any token is placed at the starting position; however, we find the cases where the token at the starting position does not trigger massive activations and attention sinks, such as Mistral-7B.

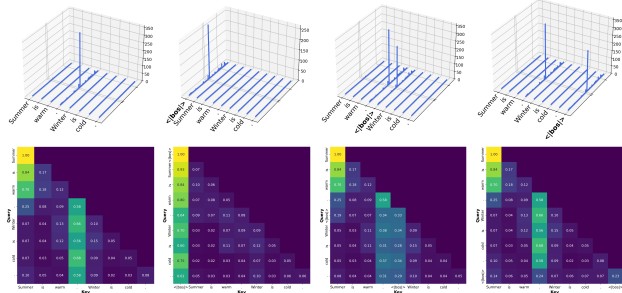

*Figure 2.* (Top) Magnitudes of the hidden state and (Bottom) attention scores after Softmax of Mistral-7B, according to the position of the *bos* token. The described hidden state is the output of layer 16 (i.e., $h_{16}$). The attention scores are calculated at layer 17 (i.e., after massive activations appear) and averaged across different heads.

Figure 2 describes the magnitudes of activations of the hidden state and normalized attention scores of Mistral-7B according to the position of the *bos* token, after massive activations appear. The reason for arbitrarily inserting the *bos* token is based on the previous observation that non-semantic tokens can trigger massive activations in certain LLMs. We use the implementation[1] of massive activations for this example and visualization. Mistral-7B has massive activations at the first delimiter token '.', not at the starting position (first column). However, when the *bos* token is placed at the starting position, it triggers massive activations and the first delimiter token loses its massive activations (second column). When the *bos* token is inserted in the middle or ending position after the first delimiter token, massive activations are observed in both tokens (third and fourth columns). On the other hand, there are models that respond to the starting position but not to the *bos* token, such as Llama-2, detailed in Appendix C. Therefore, by considering both conditions, we use only the *bos* token placed at the starting position for the continuation of analysis and algorithm development.

**Massive activations originate in the intermediate state of a FFN module.** Next, we trace various states in early layers until the first massive activations appear, using the *bos* token. Specifically, we monitor $h_{l-1}$, $\text{LN}(h_{l-1})$, $\text{ATTN}(\text{LN}(h_{l-1}))$, $\hat{h}_l$, $\text{LN}(\hat{h}_l)$, and $\text{FFN}(\text{LN}(\hat{h}_l))$ in Eq. (1) throughout early layers. Figure 3(a) illustrates the magnitudes of various states in layers $l \in [1, 3]$. It is observed that $\text{FFN}(\text{LN}(\hat{h}_2))$ has massive activations before $h_2$. With Figure 3(b), which describes the top three and median magnitudes of the hidden state[1], it is observed that the massive activations generated within a FFN module at layer 2 are transmitted directly to the next hidden state and then propagated solely through the residual connections.

---

[1]https://github.com/locuslab/massive-activations

*Table 1.* Perplexity and zero-shot downstream tasks performance according to the attack.

| Models | WikiText | C4 | PG-19 | Avg. ($\downarrow$) | ARC-E | ARC-C | BoolQ | PIQA | WG | Avg. ($\uparrow$) |
|---|---|---|---|---|---|---|---|---|---|---|
| Llama-3-8B | 5.75 | 9.94 | 8.98 | 8.22 | 77.4 | 52.7 | 81.4 | 80.8 | 72.9 | 73.0 |
| top-5 zeroing | 104.57 | 132.83 | 130.83 | 122.74 | 29.1 | 22.0 | 41.8 | 53.8 | 50.3 | 39.4 |
| top-5 retaining | 10.15 | 23.98 | 28.72 | 20.95 | 75.0 | 48.0 | 80.2 | 77.7 | 74.1 | 71.0 |
| Llama-3-70B | 2.68 | 7.59 | 6.02 | 5.43 | 85.8 | 64.2 | 85.3 | 84.6 | 80.5 | 80.1 |
| top-5 zeroing | 11135.81 | 7288.86 | 4696.49 | 7707.05 | 28.6 | 22.8 | 38.0 | 55.5 | 49.5 | 38.9 |
| top-5 retaining | 3.47 | 9.93 | 7.26 | 6.89 | 45.9 | 25.6 | 83.9 | 64.6 | 77.0 | 59.4 |
| Llama-3.1-405B (8bit) | 1.41 | 6.20 | 3.23 | 3.61 | 86.3 | 66.0 | 88.2 | 85.0 | 81.1 | 81.3 |
| top-5 zeroing | 1785.56 | 985.36 | 633.40 | 1134.77 | 26.6 | 25.9 | 37.8 | 49.7 | 50.3 | 38.1 |
| top-5 retaining | 2.47 | 9.36 | 5.61 | 5.81 | 83.0 | 62.5 | 83.1 | 83.2 | 70.3 | 76.4 |

(a) Various states in early layers.

(b) Hidden states.

(c) Intermediate states.

*Figure 3.* (a) Magnitudes of various states and (b and c) the top three and median magnitudes of hidden states and intermediate states across layers. These results show that massive activations in the hidden state originate from those in the intermediate state in a FFN module in an early layer.

Furthermore, we decompose a FFN module into $\boldsymbol{W}_{down}(\sigma(\boldsymbol{W}_{gate}(\cdot)) \odot \boldsymbol{W}_{up}(\cdot))$, to analyze the intermediate states (i.e., the output of $\sigma(\boldsymbol{W}_{gate}(\cdot)) \odot \boldsymbol{W}_{up}(\cdot)$). Figure 3(c) describes the top three and median magnitudes of the intermediate state across layers. It is demonstrated that *massive activations originate in the intermediate state of a FFN module in an early layer $l$*. This result implies that $\boldsymbol{W}_{up}$ and $\boldsymbol{W}_{gate}$ in layer $l$ are closely tied to massive activations. Additional results for other LLMs are provided in the Appendix D.

### 2.3. Massive Weights

Massive weights are defined based on massive activations in the intermediate state at layer $l$, denoted as $\hat{h}_l^{inter}$, when the *bos* token is fed into LLMs. To elaborate, we define the rows in the projection matrix $\boldsymbol{W}_{up}$ (and $\boldsymbol{W}_{gate}$, if it exists) that correspond to the indices of the top-$k$ magnitudes in $\hat{h}_l^{inter}$ as *top-k massive weights*, depicted in Figure 1(a). It is noted that massive weights are defined within one specific layer, which means the number of massive weights is significantly smaller compared to the total number of parameters in LLMs. For example, in Llama-3-8B, the number of top-$k$ massive weights is calculated as $2 \times k \times 4096$, where 4096

represents the dimensions of hidden state. If $k$ is set to 5, massive weights account for approximately 0.0005% of the total parameters in Llama-3-8B, approximately 0.0001% in Llama-3-70B, and approximately 0.00004% in Llama-3.1-405B.

**Massive weights are extremely small in quantity, their impact is tremendous.** To assess the significance of massive weights, we conduct two types of attacks: top-$k$ zeroing and top-$k$ retaining. Note that these attacks only affect the $\boldsymbol{W}_{up}$ and $\boldsymbol{W}_{gate}$ projection matrices in layer $l$, where massive weights are present. The first attack is to set the top-$k$ massive weights to zero (i.e., darker orange weights in Figure 1(a)). In essence, this attack is very similar to the one proposed in Sun et al. (2024a), where massive activations in the hidden state are zeroed out in a single layer. The difference is that their attack targets the hidden state, while our attack targets the intermediate state. The second attack is to set all weights to zero except for top-$k$ massive weights (i.e., lighter orange weights in Figure 1(a)). That is, the number of rows being damaged in each attack is $k$ and the dimensions of intermediate state $- k$, respectively.

Following Sun et al. (2024a), we assess perplexity[1] on three datasets: WikiText (Merity et al., 2017), C4 (Raffel et al., 2020), and PG-19 (Rae et al., 2020). Additionally, we evaluate zero-shot accuracy[2] on five tasks: Arc-Easy (ARC-E), Arc-Challenge (ARC-C) (Clark et al., 2018), BoolQ (Clark et al., 2019), PIQA (Bisk et al., 2020), and WinoGrande (WG) (Sakaguchi et al., 2021). Table 1 presents the results of two attacks on Llama-3-8B, Llama-3-70B, and Llama-3.1-405B (8bit)[3], when $k$ is set to 5. Llama-3-8B has its massive weights in layer 2 out of 32 layers, Llama-3-70B in layer 4 out of 80 layers, and Llama-3.1-405B in layer 6 out of 126 layers. Detailed indices of massive weights are provided in Appendix B, including various LLMs. The larger the difference from the original performance, the stronger the attack.

---

[2]https://github.com/EleutherAI/lm-evaluation-harness

[3]We use 8xA100-80G GPUs for our work; therefore, we employ an 8-bit model.

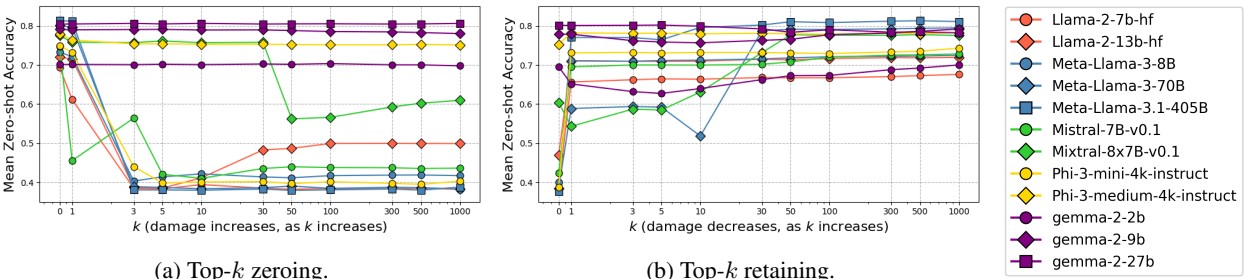

(a) Top-$k$ zeroing.  (b) Top-$k$ retaining.

*Figure 4.* Mean zero-shot accuracy of top-$k$ zeroing and retaining across various LLMs.

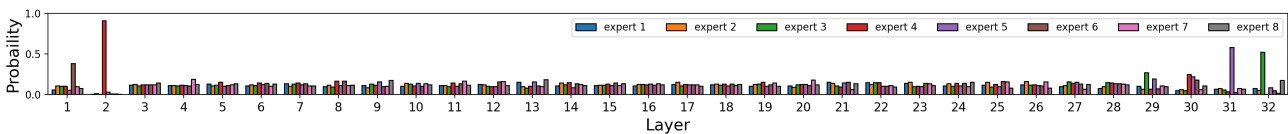

*Figure 5.* Probability of experts of Mixtral-8x7b for the *bos* token. In the layer with massive weights (i.e., layer 2), the router probability between experts becomes completely skewed to one expert.

Top-5 zeroing is a much stronger attack than top-5 retaining, even though top-5 retaining sets several thousands times more weights to zero compared to top-5 zeroing does for the same projection matrices. This means that in the projections $W_{up}$ and $W_{gate}$ at layer $l$, having only massive weights is significantly better than having all weights except for massive weights. In detail, similar to the findings of Sun et al. (2024a), the top-$k$ zeroing attack proves to be highly effective in disrupting the Llama-3 family, even for extremely large-scale models (e.g., 70B and 405B). On the other hand, the top-$k$ retaining attack does not cause complete damage. In conclusion, these findings reveal that massive weights are predominantly learned during pre-training, highlighting their essential contribution to the performance of LLMs.

Moreover, massive weights also exist in instruction-tuned LLMs such as Llama-3-8B-Instruct. These attacks are effective, as depicted in Figure 1(b). When massive weights are set to zero (i.e., darker orange box), the model repeats the same text as the user prompt. On the other hand, when all weights are set to zero except for massive weights (i.e., lighter orange box), the model retains its ability to generate text, although the generated text differs from the original.

**The value of $k$, which affects performance, varies depending on the model architecture.** We examine the robustness of various LLMs against the top-$k$ zeroing and retaining attacks, with a focus on the impact of the parameter $k$. Llama-2 (Touvron et al., 2023), Llama-3 (Dubey et al., 2024), Mistral (Jiang et al., 2023) and Mixtral (Jiang et al., 2024), Phi-3 (Abdin et al., 2024), and Gemma-2 (Team et al., 2024) families are used. Figure 4 illustrates the mean zero-shot accuracy of LLMs under the two attacks, according to the $k$. In top-$k$ zeroing, more weights are set to zero as $k$ increases, whereas in top-$k$ retaining, more weights are

set to zero as $k$ decreases. When $k$ is 0 in top-$k$ zeroing, it corresponds to the original performance without any attack, whereas, when $k$ is 0 in top-$k$ retaining, it sets the entire weights of $W_{up}$ and $W_{gate}$ in layer $l$ to zero.

The Llama families are highly sensitive to massive weights. In the top-$k$ zeroing, a noticeable performance drop occurs even when $k$ is as small as 3, irrespective of the model's scale. In top-$k$ retaining, when $k$ is set to 1 (i.e., with only one row active in $W_{up}$ and $W_{gate}$ in layer $l$), the performance nearly reaches the original level in smaller-scaled models ($\leq$ 13B). While, for larger-scaled models ($\geq$ 70B), the top-30 massive weights are required to maintain performance. Moreover, since different versions of Llama families pre-trained on varying datasets (e.g., with different cutoff dates) exhibit similar tendencies, it can be inferred that the pre-training dataset itself does not significantly influence the emergence of massive phenomena.

Similarly, Mistral is also significantly disrupted, when $k$ is set to 5. Mixtral is a sparse Mixture of Experts (MoE) architecture that uses a top-2 routing mechanism, where two experts are activated among eight FFN modules in each layer. To attack the Mixtral model, we identify the active experts in the layer with massive weights using the *bos* token. Figure 5 describes the probability distribution of experts in the Mixtral model across all layers. Notably, it is observed that when massive activations occur, a single expert (i.e., expert 4) is assigned a significantly higher probability than the others. Therefore, we target only the $W_{up}$ and $W_{gate}$ of this expert, rather than all experts. Although Mixtral does not completely break down, there is a considerable decline in performance when the top-50 massive weights are zeroed out. These results indicate that, despite the immense resources required to build high-performance LLMs, they can collapse like a house of cards even under minimal attacks.

The Phi-3 family exhibits different robustness against attacks depending on the model size. As noted by Abdin et al. (2024), the phi-3-mini (3.8B) is trained on 3.3T tokens, while the phi-3-medium (14B) is trained on 4.8T tokens. A key architectural difference from the Llama family is the use of dropout to the outputs of both the ATTN and FFN modules, formed by Eq. (2). While a specific recipe for dropout is not provided in the technical report (Abdin et al., 2024), in the case of phi-3-medium, applying dropout with longer pre-training might ensure that the residual connections contribute meaningfully, mitigating the risk of excessive dependence on massive weights.

$$h_l = \hat{h}_l + \text{dropout}(\text{FFN}(\text{LN}(\hat{h}_l))),$$
$$\text{where } \hat{h}_l = h_{l-1} + \text{dropout}(\text{ATTN}(\text{LN}(h_{l-1}))) \tag{2}$$

The Gemma-2 family is exceptionally resilient to the top-$k$ zeroing attack, maintaining almost no loss in performance even when $k$ is large. Additionally, even if $\boldsymbol{W}_{up}$ and $\boldsymbol{W}_{gate}$ are entirely eliminated (i.e., $k = 0$ in top-$k$ retaining), there is no noticeable performance degradation. This family incorporates two additional LN layers after both the ATTN and FFN modules, formed by Eq. (3). These added normalization layers result in completely different hidden and intermediate states compared to other LLMs, as described in Appendix D. Furthermore, attention sinks are not observed in the Gemma-2 family, as shown in Appendix E.

$$h_l = \hat{h}_l + \text{LN}(\text{FFN}(\text{LN}(\hat{h}_l))),$$
$$\text{where } \hat{h}_l = h_{l-1} + \text{LN}(\text{ATTN}(\text{LN}(h_{l-1}))) \tag{3}$$

These observations imply that LLMs not sensitive to massive phenomena may not perform effectively with algorithms based on massive phenomena. Further discussion is provided in Appendix F.

## 3. Massive Weights Curriculum Dropout

In this section, we propose a straightforward plug-and-play method, termed **ma**ssive weights **c**urriculum **drop**out (MacDrop), during parameter-efficient fine-tuning such as low rank adaptation (Hu et al., 2022). This method applies dropout to the pre-trained massive weights with a curriculum that gradually reduces the dropout probability. The reason for applying dropout to weights (Wan et al., 2013) instead of activations (Srivastava et al., 2014) is that the number of massive activations is only $k$, but that of massive weights is $k \times d$, where $d$ is the dimension of the hidden states. It is important to note that our method does not modify the trainable parameters of adapters; instead, it is applied to the pre-trained frozen weights. Therefore, MacDrop can be applied orthogonally to the process of training the adapter.

---

**Algorithm 1:** Top-$k$ massive weights curriculum dropout (MacDrop) in pseudo PyTorch style

```
// Dropout is only executed in layer l,
   where h_l^inter appears.
```
**Input:** $k$, massive intermediate state $h_l^{inter}$ of the *bos* token, initial dropout probability $p_0$, total steps $T$
```
// massive activations in the
   intermediate state
```
1   _, sorted_indices = torch.sort(torch.abs($h_l^{inter}$), descending=True)
2   massive_indices = sorted_indices[:$k$]
```
   // pre-trained massive weights
```
3   massive_W_up = copy(W_up[massive_indices, :])
4   massive_W_gate = copy(W_gate[massive_indices, :])
5   **for** $t = 1$ *to* $T$ **do**
```
      // decreasing dropout probability
```
6     $p = p_0 \times (1 - \frac{t}{T})$
```
      // pre-trained massive weights
         dropout
```
7     mask = (torch.rand(massive_W_up.shape) > $p$).int()
8     W_up[massive_indices, :] *= mask
9     W_gate[massive_indices, :] *= mask
10    tr_loss_step = training_step(model, inputs)
```
      // pre-trained massive weights
         rollback
```
11    W_up[massive_indices, :] = massive_W_up
12    W_gate[massive_indices, :] = massive_W_gate

---

MacDrop is motivated by the observation that massive weights are predominantly learned during pre-training, and that zeroing them can severely undermine LLMs. Therefore, at the early stages of fine-tuning, the objective is to reduce the reliance on massive weights, as their excessive dominance may lead the model to over-rely on specific patterns. Moreover, considering that the undamaged pre-trained model is used after fine-tuning is finished, we develop a strategy to adjust the dropout rate using a curriculum.

Algorithm 1 describes MacDrop in a pseudo PyTorch (Paszke et al., 2019) style, and is implemented within the trainer code of transformers[4]. Initially, massive weights are identified using the *bos* token before fine-tuning (Lines 1-4). Subsequently, an adapter is trained while the pre-trained massive weights are dropped. Meanwhile, a curriculum strategy is applied to progressively enable the use of the original pre-trained weights without masking. Note that in Line 10, 'model' includes both the masked pre-trained network and a trainable adapter. When we implement Lines 1-4 in practice, we precomputed the massive indices and loaded them. Lines 6-9 and 11-12 are the additional computation for MacDrop, which requires neglectable overhead (e.g., approximately 0.35 second per step for Llama-3-8B using LoRA on 8xA100 GPUs).

---

[4]https://github.com/huggingface/transformers/tree/main/src/transformers

*Table 2.* Long context tasks performance.

| Method | Single-document QA | | | Multi-document QA | | | Summarization | | | Few-shot learning | | | Synthetic | | Code | | Avg. |
|---|---|---|---|---|---|---|---|---|---|---|---|---|---|---|---|---|---|
| | NrtvQA | Qasper | MF-en | HotpotQA | 2WikiMQA | Musique | GovReport | QMSum | MultiNews | TREC | TriviaQA | SAMSum | PCount | PRe | Lcc | RB-P | |
| Model: Meta-Llama-3-8B | | | | | | | | | | | | | | | | | |
| LoRA | 26.03 | 30.38 | 53.38 | 26.30 | 23.05 | 11.96 | 29.00 | 22.81 | 26.43 | 72.50 | 81.14 | 44.27 | 2.63 | 32.00 | 72.90 | 69.70 | 39.03 |
| + MacDrop | 25.31 | 34.05 | 46.84 | 38.06 | 28.99 | 17.92 | 29.62 | 22.86 | 26.64 | 72.00 | 89.34 | 45.08 | 3.00 | 27.50 | 73.17 | 68.25 | **40.54** |
| DoRA | 26.31 | 31.57 | 52.10 | 27.04 | 23.57 | 12.01 | 29.20 | 23.35 | 26.34 | 73.50 | 81.35 | 43.22 | 2.61 | 30.00 | 73.46 | 69.44 | 39.07 |
| + MacDrop | 26.11 | 30.99 | 53.37 | 29.35 | 25.66 | 12.14 | 29.07 | 23.10 | 26.32 | 73.50 | 86.63 | 44.64 | 2.05 | 25.50 | 73.90 | 69.32 | **39.48** |
| Model: Mistral-7B-v0.1 | | | | | | | | | | | | | | | | | |
| LoRA | 22.52 | 34.64 | 35.65 | 32.11 | 19.80 | 12.73 | 27.22 | 21.98 | 26.73 | 69.00 | 87.50 | 41.70 | 1.00 | 21.00 | 71.57 | 65.47 | 36.91 |
| + MacDrop | 23.49 | 38.51 | 36.11 | 37.78 | 27.60 | 14.91 | 26.40 | 22.53 | 26.92 | 69.50 | 89.92 | 37.07 | 1.55 | 20.00 | 70.37 | 65.87 | **38.03** |
| DoRA | 22.79 | 34.52 | 35.55 | 30.87 | 17.84 | 12.26 | 27.45 | 22.15 | 26.52 | 70.00 | 88.05 | 41.56 | 1.00 | 20.50 | 71.88 | 65.38 | 36.77 |
| + MacDrop | 23.10 | 35.10 | 35.00 | 29.53 | 23.77 | 10.50 | 27.14 | 22.63 | 27.50 | 69.00 | 89.56 | 38.96 | 1.00 | 21.42 | 71.17 | 65.53 | **36.93** |

*Table 3.* Zero-shot downstream tasks performance.

| Model | Method | ARC-E | ARC-C | BoolQ | PIQA | WG | Avg. |
|---|---|---|---|---|---|---|---|
| Llama-3-8B | LoRA | 79.6 | 58.2 | 83.9 | 82.4 | 75.9 | 76.0 |
| | + MacDrop | 82.9 | 58.3 | 83.9 | 82.6 | 75.0 | **76.5** |
| | DoRA | 80.8 | 57.7 | 83.9 | 82.5 | 75.8 | 76.1 |
| | + MacDrop | 81.9 | 58.2 | 83.9 | 82.2 | 75.6 | **76.4** |
| Mistral-7B | LoRA | 78.5 | 54.9 | 84.9 | 82.9 | 75.3 | 75.3 |
| | + MacDrop | 80.9 | 56.7 | 85.0 | 83.0 | 75.3 | **76.2** |
| | DoRA | 78.4 | 55.1 | 85.0 | 82.9 | 75.1 | 75.3 |
| | + MacDrop | 80.6 | 56.7 | 85.3 | 82.9 | 75.1 | **76.1** |

*Table 4.* Zero-shot downstream tasks performance under the top-3 zeroing attack, related to robustness.

| Model | Method | ARC-E | ARC-C | BoolQ | PIQA | WG | Avg. |
|---|---|---|---|---|---|---|---|
| Llama-3-8B | LoRA | 29.9 | 22.9 | 45.7 | 52.9 | 50.7 | 40.4 |
| | + MacDrop | 36.8 | 24.7 | 64.3 | 58.5 | 54.1 | **47.7** |
| | DoRA | 29.8 | 23.0 | 46.0 | 52.6 | 50.0 | 40.3 |
| | + MacDrop | 78.7 | 53.7 | 79.9 | 79.0 | 72.4 | **72.7** |
| Mistral-7B | LoRA | 54.6 | 34.8 | 58.0 | 74.8 | 57.5 | 55.9 |
| | + MacDrop | 69.2 | 45.0 | 78.3 | 79.3 | 64.1 | **67.2** |
| | DoRA | 55.7 | 35.3 | 58.6 | 74.4 | 58.2 | 56.4 |
| | + MacDrop | 78.1 | 52.4 | 84.1 | 82.3 | 69.5 | **73.3** |

# 4. Experiments

## 4.1. Zero-shot Downstream Task

We fine-tune the Llama-3-8B and Mistral-7B using the alpaca_gpt4_en dataset (Peng et al., 2023) for 3 epochs (579 steps), and evaluate on five zero-shot tasks. We use two parameter-efficient fine-tuning (PEFT) methods, LoRA (Hu et al., 2022) and DoRA (yang Liu et al., 2024). DoRA decomposes the pre-trained weights into two components, magnitude and direction, and applies LoRA to the direction component. Our method is based on the implementation of Llama-Factory (Zheng et al., 2024)[5]. For MacDrop, $k$ and $p_0$ are set to 5 and 0.8, respectively. Details for implementations are explained in Appendix A.

Table 3 presents the results on zero-shot downstream tasks. For both the models and methods, MacDrop consistently leads to performance gains, especially in ARC-Easy and ARC-Challenge tasks. Note that we do not claim MacDrop is superior to LoRA/DoRA; rather, we demonstrate that LoRA/DoRA performs better when MacDrop is applied than when it is not. Results for other LLMs are provided in Appendix F.

MacDrop is designed to mitigate dependence on massive weights during PEFT. To assess whether this dependency is effectively reduced, we attack the previous fine-tuned models in Table 3. Table 4 shows the performance changes whether applying MacDrop under the top-3 zeroing attack. The zeroing attack severely degrades the performance of fine-tuned models without MacDrop, as seen in the case of Llama-3-8B with DoRA, where performance drops from 76.1 to 40.3. However, fine-tuned models with MacDrop exhibit significantly better performance under attack, indicating better robustness. Especially, when MacDrop is combined with DoRA, it demonstrates remarkable robustness. For instance, in the case of Llama-3-8B with DoRA, where performance drops from 76.4 to 72.7.

## 4.2. Long Context Task

We evaluate the two LLMs on LongBench (Bai et al., 2024), a benchmark specifically designed to assess the ability to understand long contexts. This includes 5 sub-categories and 16 English datasets: single-document QA, multi-document QA, summarization, few-shot learning, synthetic, and code generation. We set the max length of models to 7,500. Table 2 shows that MacDrop increases the performance when understanding long context is required.

---

[5]https://github.com/hiyouga/LLaMA-Factory

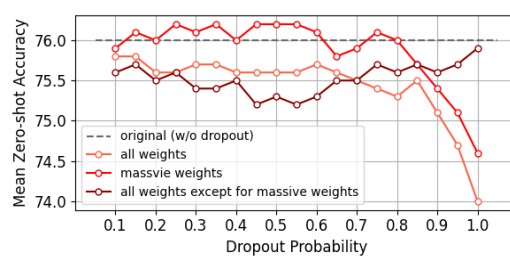

*Figure 6.* Mean zero-shot accuracy according to the dropout scope and probability $p$.

### 4.3. Ablation Study

We further provide ablation studies related to `MacDrop` using Llama-3-8B. The variations used in this section can serve as a baseline.

#### 4.3.1. DROPOUT SCOPE AND PROBABILITY

We investigate the effect of dropout scope and probability compared to the original performance achieved through LoRA without dropout. This ablation study is also conducted on the $W_{up}$ and $W_{gate}$ projection matrices in layer $l$. The dropout scope is divided into three categories: all weights, massive weights, all weights except for massive weights. Additionally, to assess the impact of dropout probability, it is kept constant throughout the fine-tuning process, without using a curriculum.

Figure 6 illustrates the mean zero-shot accuracy according to the dropout scope and dropout probability $p$. It is observed that among three scopes, the original performance (i.e., without dropout), represented by the dotted line at 76.0, can be surpassed only when dropout is applied solely to massive weights. Nevertheless, if strong dropout (e.g., $p \geq 0.85$) is maintained on the pre-trained massive weights during fine-tuning, performance deteriorates. This highlights the need to safeguard the pre-trained massive weights during the final stages of fine-tuning, because we are ultimately using them without causing any damage.

#### 4.3.2. CURRICULUM METHODS AND INITIAL DROPOUT PROBABILITY

We investigate the effect of curriculum methods and initial dropout probability $p_0$ in `MacDrop`, when LoRA is applied. We compare four curriculum methods: step-wise linear (Step), before epoch-wise linear (Epoch(before)), after epoch-wise linear (Epoch(after)), and exponential (Exp.). In formula, Step is defined as $p = p_0 \times (1 - \frac{t_{step}}{T_{step}})$. Epoch(before) and Epoch(after) are defined as $p = p_0 \times (1 - \frac{t_{epoch}-1}{T_{epoch}})$ and $p = p_0 \times (1 - \frac{t_{epoch}}{T_{epoch}})$, respectively. Exp. is defined as $p = p_0 \times exp(-\alpha t_{step})$. Figure 7 describes dropout probability $p$ according to curriculum methods, when $p_0$ is 1.0. The distinct difference between Epoch(before) and Epoch(after) is that at the final epoch, the former continues

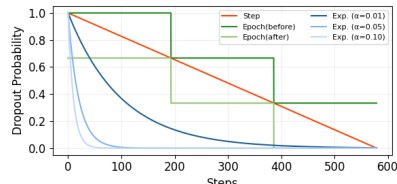

*Figure 7.* Curriculum methods.

*Table 5.* Mean zero-shot accuracy according to curriculum methods and initial dropout probability.

| Curriculum Method | $p_0$ | | | |
|---|---|---|---|---|
| | 0.2 | 0.5 | 0.8 | 1.0 |
| Step | 76.0 | 76.3 | 76.5 | 75.5 |
| Epoch(before) | 76.1 | 76.1 | 76.1 | 75.5 |
| Epoch(after) | 75.9 | 76.0 | 76.2 | 76.3 |
| Exp. ($\alpha = 0.01$) | 76.0 | 76.2 | 76.5 | 76.4 |
| Exp. ($\alpha = 0.05$) | 76.0 | 76.1 | 76.2 | 76.3 |
| Exp. ($\alpha = 0.10$) | 76.0 | 76.1 | 76.1 | 76.2 |
| Mean | 76.0 | 76.1 | 76.3 | 76.0 |

to apply dropout to the pre-trained massive weights with a probability of $p_0 \times \frac{1}{T_{epoch}}$, while the latter fully utilizes the pre-trained massive weights.

Table 5 presents mean zero-shot accuracy according to curriculum methods and initial dropout probability $p_0$. It is observed that step-based curriculum methods (e.g., Step and Exp.) generally achieve greater performance improvements compared to epoch-based curriculum methods (e.g., Epoch(before) and Epoch(after)). Nevertheless, when the initial dropout probability is relatively low (e.g., $p_0 \leq 0.2$), even step-based curriculum methods fail to bring any performance gain compared to the original performance of 76.0. Additionally, it is shown that using a smaller $\alpha$ in the Exp. method leads to greater performance improvements, suggesting that a rapid decline in dropout probability to zero can diminish the effectiveness of `MacDrop`. On the other hand, for the Step and Epoch(before) methods, a significant performance drop is observed at a $p_0$ value of 1.0, highlighting the necessity of a near-zero dropout probability for the end of fine-tuning. In conclusion, for `MacDrop`, we recommend using the Step or Exp. with a smaller $\alpha$, initiated from a moderately high $p_0$.

## 5. Conclusion

In this paper, we explore the weight space of LLMs and identify the presence of massive weights within a FFN module in an early layer, which are predominantly pre-trained and have a significant impact on the performance of LLMs. Based on our observation, we propose a plug-and-play fine-tuning method called `MacDrop`, which applies dropout to the pre-trained massive weights, rather than to the parameters of adapters, during parameter-efficient fine-tuning. We hope that our findings will inspire future research in weight space of LLMs.

## Impact Statement

This paper seeks to contribute to the advancement of large language models, particularly by providing deeper insights into their internal mechanisms. Our research demonstrates, through various open-sourced LLMs, that in many cases, models can be easily disrupted by zeroing massive weights if access to their weights is available. Subsequently, we improve robustness against such attacks through `MacDrop`. The existence of massive weights does not inherently present a risk if the model weights remain undisclosed. However, we believe that model developers should understand these internal mechanisms and take responsibility for ensuring robustness. In this vein, our work has the potential to make a meaningful impact.

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

# A. Implementation Details

We use 8xA100-80GB, for our all implementations. As discussed in Section 2.2, we use only the *bos* token to analyze massive activations and massive weights, and design `MacDrop`. In Section 2.2, we use the example and visualization of massive activations. For parameter-efficient fine-tuning, Llama-Factory is used and configurations are summarized in Table 6. For evaluation, we use the code of massive activations for perplexity, use lm-eval-harness for zero-shot accuracy, and use FastChat and Prometheus for generation tasks. Related papers and codes are cited in the main.

*Table 6.* Configuration for low rank adaptation (LoRA and DoRA).

| Argument | Setting |
|---|---|
| dataset | alpaca_gpt4_en |
| validation size | 0.05 |
| per device train batch size | 8 |
| gradient accumulation steps | 4 |
| learning rate | 1e-4 |
| num train epochs | 3 |
| warmup ratio | 0.05 |
| adam $\beta_1$ | 0.9 |
| adam $\beta_2$ | 0.999 |
| lora target | all linear layers except for embedding layer and lm head |
| lora rank | 16 |
| lora alpha | 16 |

# B. Position of Massive Weights

Table 7 summarizes the position of massive weights across various models. These are selected based on the magnitudes of intermediate state in Appendix D.

*Table 7.* Layer and indices of top-5 massive weights.

| Model | Layer | Top-5 indices |
|---|---|---|
| Llama-2-7b-hf | 2 | [7890, 10411, 1192, 8731, 5843] |
| Llama-2-7b-chat-hf | 2 | [7890, 10411, 1192, 8731, 5843] |
| Llama-2-13b-hf | 4 | [7678, 8811, 11371, 6619, 12281] |
| Llama-2-13b-chat-hf | 4 | [7678, 8811, 11371, 6619, 12281] |
| Meta-Llama-3-8B | 2 | [2427, 198, 6412, 12657, 591] |
| Meta-Llama-3-8B-Instruct | 2 | [2427, 198, 6412, 591, 12657] |
| Meta-Llama-3-70B | 4 | [16581, 3590, 16039, 19670, 13266] |
| Meta-Llama-3-70B-Instruct | 4 | [16581, 3590, 16039, 19670, 13266] |
| Meta-Llama-3.1-405B (8bit) | 6 | [11891, 30740, 2392, 36238, 12328] |
| Meta-Llama-3.1-405B-Instruct (8bit) | 6 | [11891, 30740, 36238, 2392, 1073] |
| Mistral-7B-v0.1 | 2 | [7310, 8572, 2514, 1878, 8693] |
| Mistral-7B-Instruct-v0.1 | 2 | [7310, 8572, 2514, 2484, 1878] |
| Mixtral-8x7B-v0.1 | 2 (expert 4) | [7310, 7530, 11981, 7492, 3178] |
| Mixtral-8x7B-Instruct-v0.1 | 2 (expert 4) | [7310, 11981, 2514, 7530, 3178] |
| Phi-3-mini-4k-instruct | 3 | [808, 340, 3644, 2473, 2987] |
| Phi-3-medium-4k-instruct | 6 | [181, 7540, 19, 15874, 5137] |
| gemma-2-2b | 2 | [1257, 2896, 6954, 8624, 7118] |
| gemma-2-2b-it | 2 | [1257, 2896, 6954, 8624, 9140] |
| gemma-2-9b | 1 | [2769, 6656, 4889, 14293, 11065] |
| gemma-2-9b-it | 1 | [2769, 6656, 4889, 14293, 10429] |
| gemma-2-27b | 10 | [34659, 32862, 9590, 8959, 32744] |
| gemma-2-27b-it | 10 | [34659, 32862, 9590, 32744, 8959] |

## C. *bos* Token Analysis for Various LLMs

In this section, we provide the magnitudes of activations of the hidden state and normalized attention scores according to the position of the *bos* token, after massive activations appear (specifically, in the middle layer), for various LLM families.

### C.1. Llama-2 family

Llama-2-7B (Figure 8) has massive activations at the starting token or first delimiter token (first column). When the *bos* token is placed in the starting position, it triggers massive activations and the 'Summer' token loses its massive activations, while first delimiter token '.' still keeps its massive activations (second column). When the *bos* token is placed in the middle or ending position after the first delimiter token, it does not trigger massive activations (third and fourth columns).

Llama-2-13B (Figure 9) has massive activations only at the starting token, other than Llama-2-7B (first column). In cases where the *bos* token is inserted, the same tendencies are observed as with the LLaMA-2-7B model.

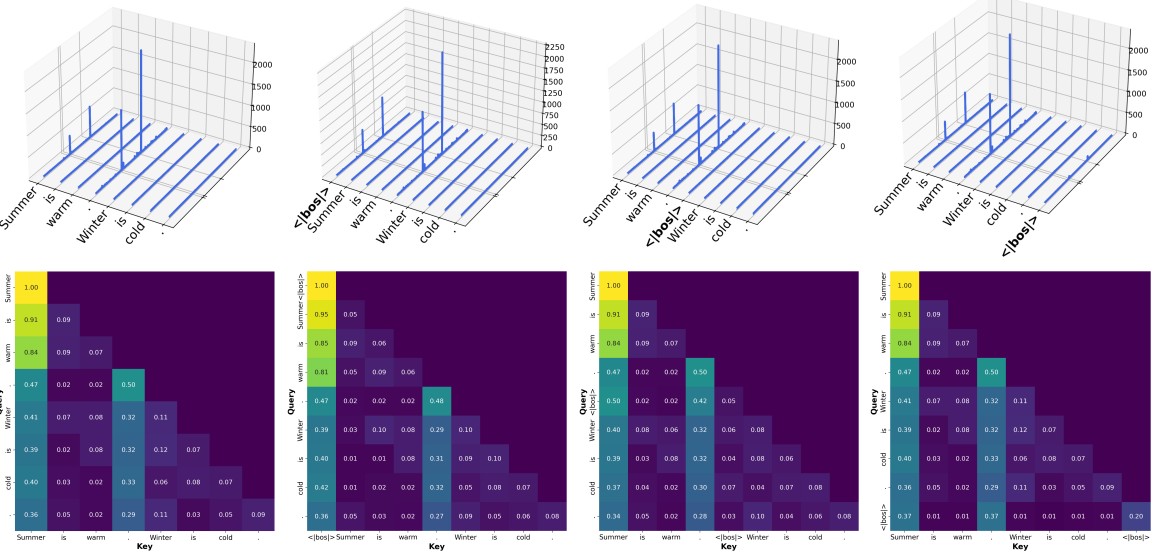

*Figure 8.* (Top) Magnitudes of the hidden state and (Bottom) attention scores of Llama-2-7b-hf.

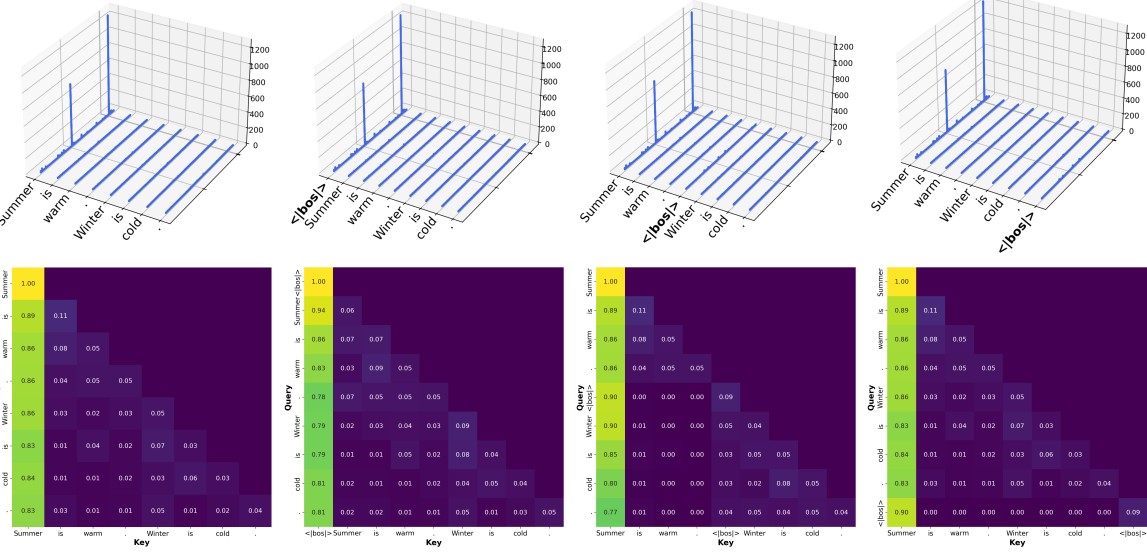

*Figure 9.* (Top) Magnitudes of the hidden state and (Bottom) attention scores of Llama-2-13b-hf.

## C.2. Llama-3 family

Llama-3-8B (Figure 10) does not have massive activations at delimiter tokens such as '.' (first column). When the *bos* token is placed in the starting position, it triggers massive activations and the 'Summer' token loses its massive activations, similar to Llama-2 family (second column). When the *bos* token is placed in the middle or ending position, it also triggers massive activations in the same feature dimensions (third and fourth columns). Namely, the *bos* token has massive activations, regardless of its position. What is intriguing is that, despite the difference in magnitude according to the position, the *bos* token similarly exhibits attention sinks.

Llama-3-70B (Figure 11) generally exhibits similar trends to Llama-3-8B. One notable difference is that the degree of sinking for the token at the first position is significantly stronger compared to that of the Llama-3-8B.

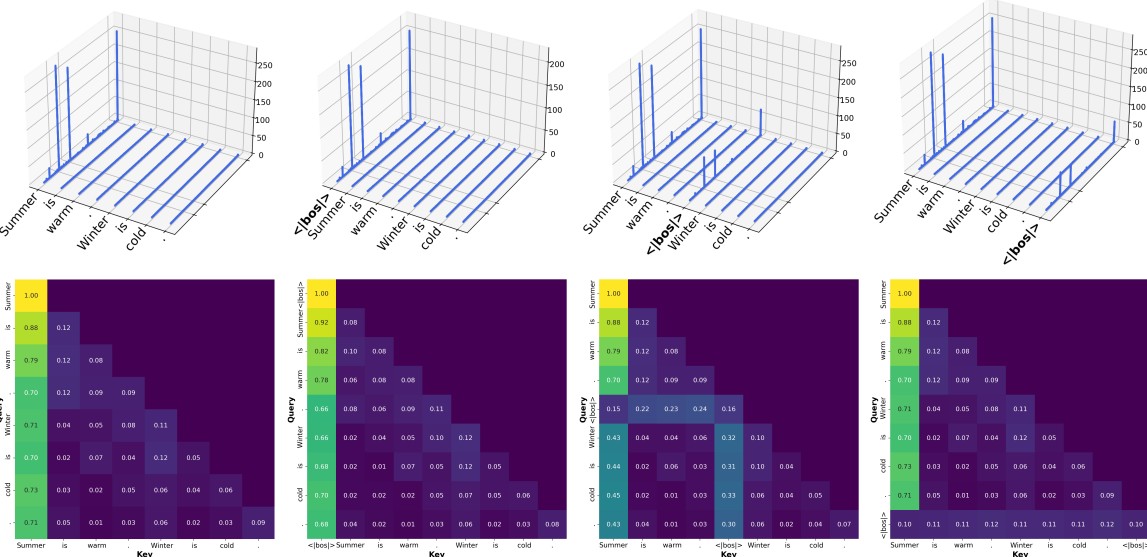

*Figure 10.* (Top) Magnitudes of the hidden state and (Bottom) attention scores of Meta-Llama-3-8B.

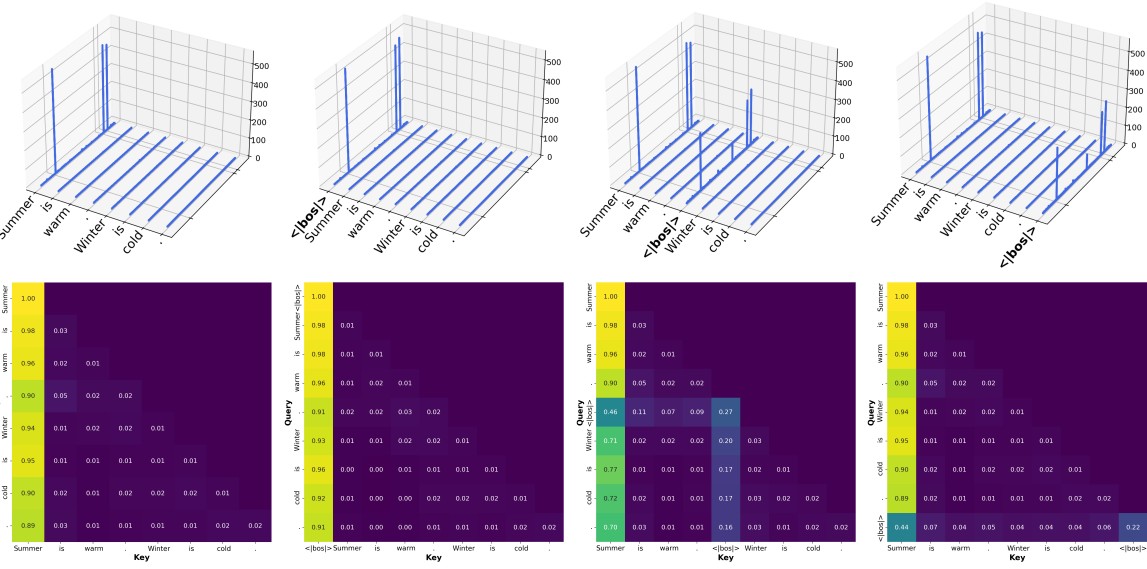

*Figure 11.* (Top) Magnitudes of the hidden state and (Bottom) attention scores of Meta-Llama-3-70B.

## C.3. Mistral and Mixtral family

Mistral (Figure 12) does not exhibit massive activations at the starting position and does not trigger attention sink, contrary to previous findings observed by Sun et al. (2024a). Rather, massive activations are observed only at the first delimiter token (first column). When the *bos* token is placed in the starting position, the first delimiter token loses its massive activations (second column). However, when the *bos* token is placed in the middle or ending position after the first delimiter token, massive activations are observed in both tokens (third and fourth columns). Similar to Llama-3 family, the *bos* token has massive activations, regardless of its position.

Mixtral (Figure 13) exhibits the same behavior as Mistral. The only difference is observed in the magnitude of its massive activations, with Mixtral producing values approximately ten times higher than Mistral.

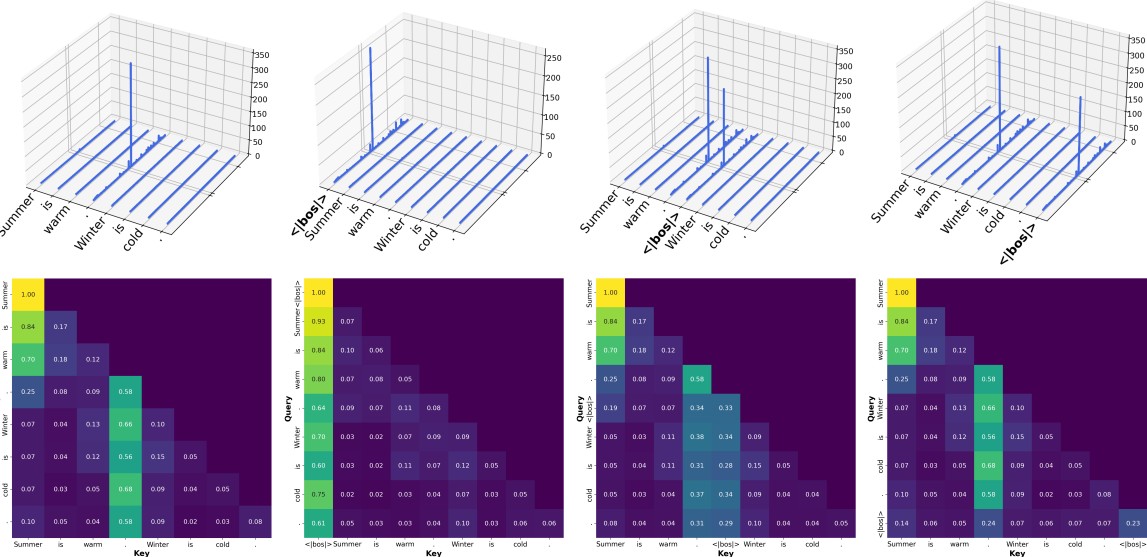

*Figure 12.* (Top) Magnitudes of the hidden state and (Bottom) attention scores of Mistral-7B-v0.1.

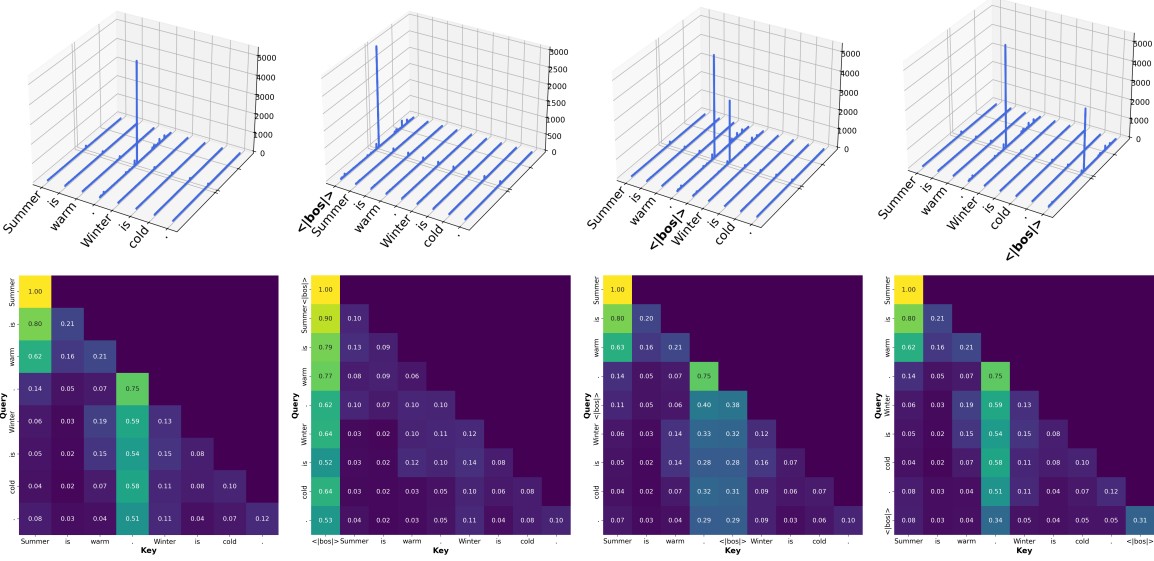

*Figure 13.* (Top) Magnitudes of the hidden state and (Bottom) attention scores of Mixtral-8x7B-v0.1.

### C.4. Phi-3 family

Phi-3-mini (Figure 14) and Phi-3-medium (Figure 15) exhibit a similar tendency to Llama-2-13B. This family has massive activations only at the starting token (first column), with a similar response when the *bos* token is inserted (second, third, and fourth column). A significant distinction between the Llama-2-13B model and the Phi-3 family lies in their attention mechanisms. Specifically, the Phi-3 family demonstrates weaker attention on the token at the first position than Llama-2-13B model. This reduced attention appears to be primarily redistributed to recent tokens.

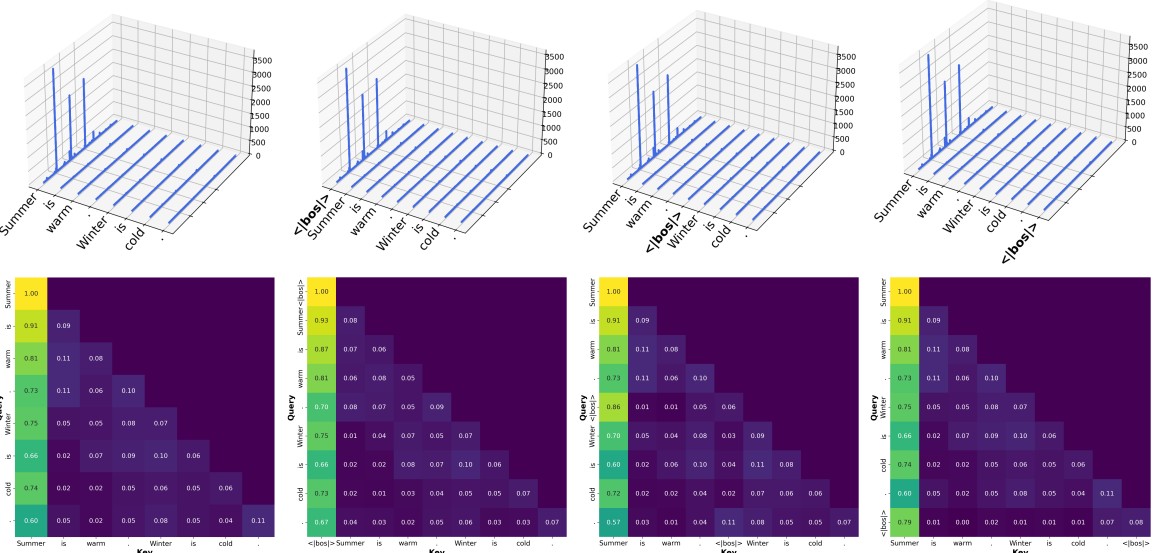

*Figure 14.* (Top) Magnitudes of the hidden state and (Bottom) attention scores of Phi-3-mini-4k-instruct.

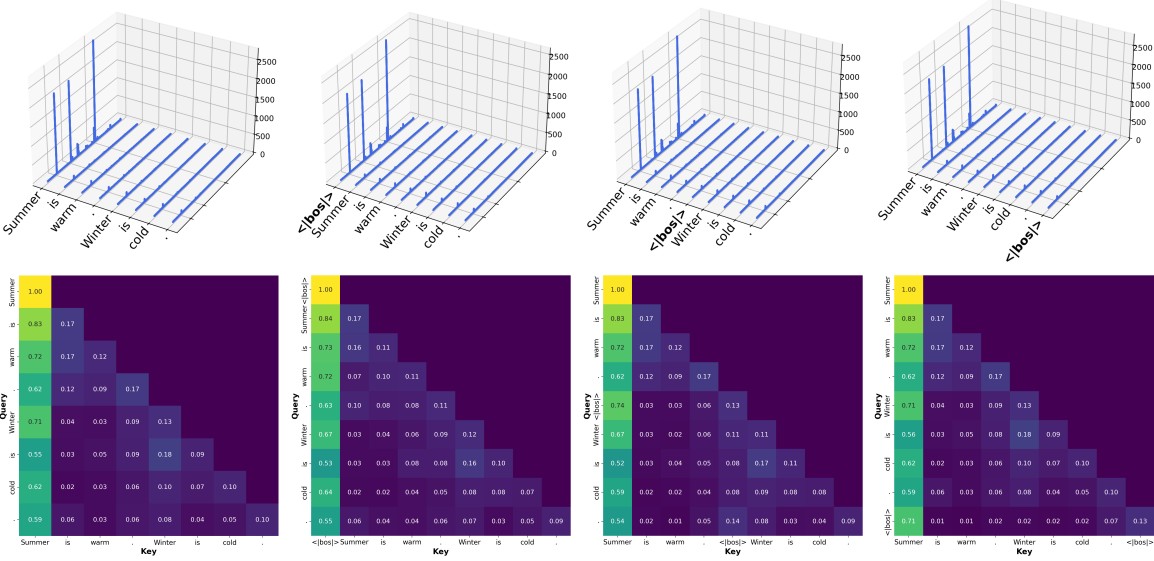

*Figure 15.* (Top) Magnitudes of the hidden state and (Bottom) attention scores of Phi-3-medium-4k-instruct.

### C.5. Gemma-2 family

The Gemma-2 family displays significantly distinct magnitudes of activations and attention scores when compared to other model families. This divergence remains consistent regardless of the absence (first column) or presence (other columns) of the *bos* token.

Gemma-2-2b (Figure 16) and Gemma-2-9b (Figure 17) do not exhibit noticeably large values along either the token axis or the feature dimension axis, from the perspective of magnitudes of activations (first column, top). This suggests that massive activations are not present. As a result, the attention mechanism avoids the attention sink phenomenon and demonstrates a strong attention on the locality of recent tokens (first column, bottom). However, when the *bos* token is fed into these models, it exhibits massive activations with extremely large values in certain feature dimensions, regardless of its position (second, third, and fourth columns). Nevertheless, compared to other models where attention sinks occur, they allocate significantly greater attention to recent tokens (especially, to its own tokens).

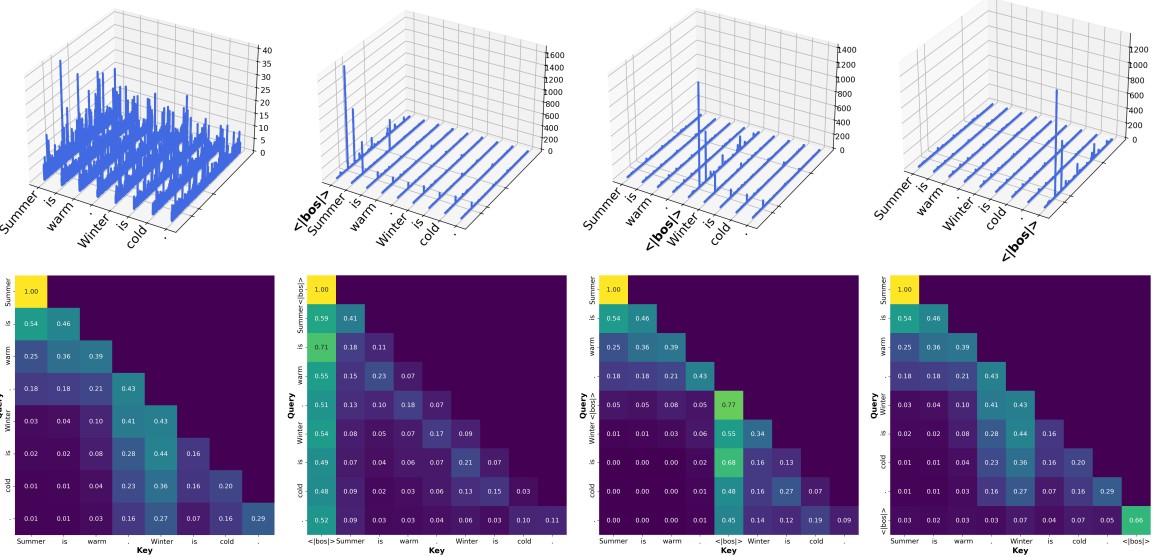

*Figure 16.* (Top) Magnitudes of the hidden state and (Bottom) attention scores of gemma-2-2b.

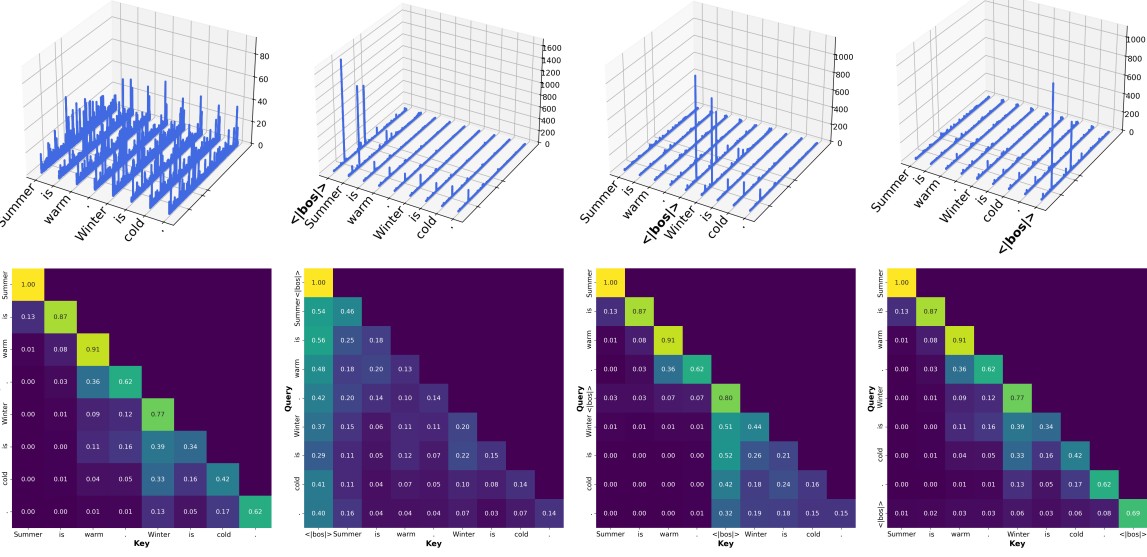

*Figure 17.* (Top) Magnitudes of the hidden state and (Bottom) attention scores of gemma-2-9b.

Gemma-2-27b (Figure 18) demonstrates a distinct behavior compared to smaller models. It exhibits noticeably large values along the feature dimension axis across all tokens, from the perspective of magnitudes of activations (first column, top). This distribution, where the differences between tokens are not pronounced, fails to create attention sinks (first column, bottom). When the *bos* token is placed in the starting position, it triggers massive activations and attention sinks by generating value that exceed the magnitudes of other tokens by more than tenfold, in the certain feature dimension (second column). However, when the *bos* token is placed in the middle or ending position, it does not trigger massive activations, similar to when the *bos* token is absent (third and fourth columns).

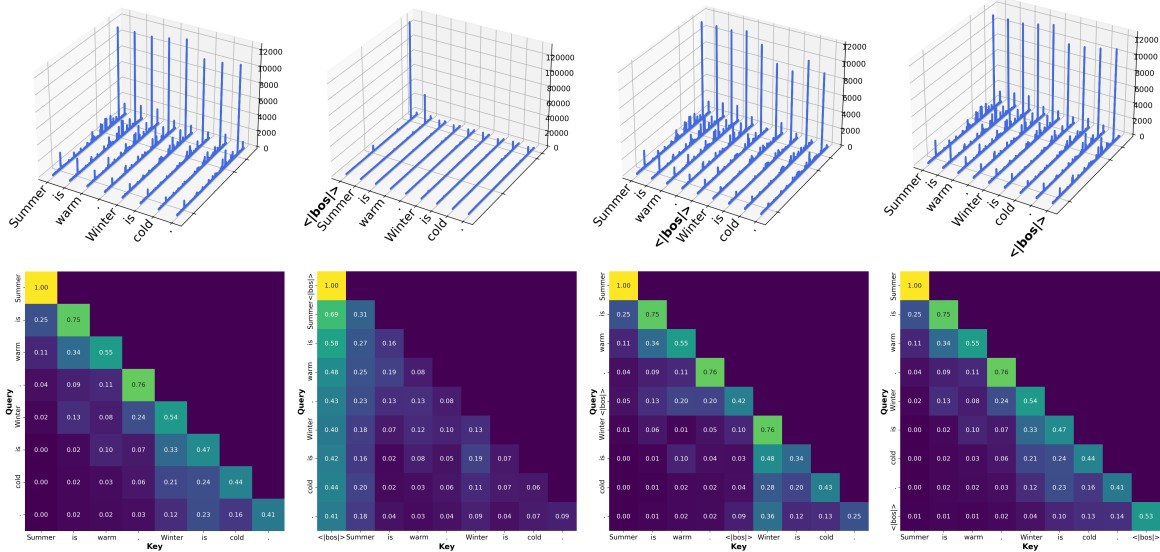

*Figure 18.* (Top) Magnitudes of the hidden state and (Bottom) attention scores of gemma-2-27b.

In summary,

- Llama-2, Llama-3, and Phi-3 families have massive activations **at the first position**.

- Llama-3, Mistral/Mixtral families, and Gemma-2-2/9b models have massive activations **at the *bos* token**.

- All families have massive activations **at the *bos* token placed at the first position**.

# D. Futher Analysis for Various LLMs

We investigate various LLM families: Llama-2 (Touvron et al., 2023), Llama-3 (Dubey et al., 2024), Mistral (Jiang et al., 2023) and Mixtral (Jiang et al., 2024), Phi-3 (Abdin et al., 2024), and Gemma-2 (Team et al., 2024). Similar to Figure 3 in the main, we provide the top-3 and median magnitudes in the hidden states and the intermediate states throughout the layers. In subcaptions, H and I represent the hidden state and the intermediate state, respectively.

When comparing pre-trained LLMs (e.g., Llama-2-7b-hf) and instruction-tuned LLMs (e.g., Llama-2-7b-chat-hf) of the same model, the shape of their graphs is almost identical. This suggests that massive weights are formed during the pre-training process. Llama-2, Llama-3, Mistral, Mixtral, and Phi-3 exhibit similar patterns in their hidden states: following a single explosive amplification in an early layer, massive activations are sustained through residual connections almost until the final layer, although Phi-3 experiences a few additional amplifications. In fact, as we discuss in the main, such an explosion initially occurs in the intermediate state, and this phenomenon is observed across different models. However, the behavior of the Gemma-2 family significantly deviates from that of other models. Firstly, instead of the values being maintained in the hidden state, Gemma-2 shows a continuous increase followed by a decrease. Secondly, the magnitude of the explosion observed in the intermediate state is considerably lower compared to other models. These unique characteristics suggest that Gemma-2 operates under different internal dynamics, which may influence its overall performance and stability.

## D.1. Llama-2 family

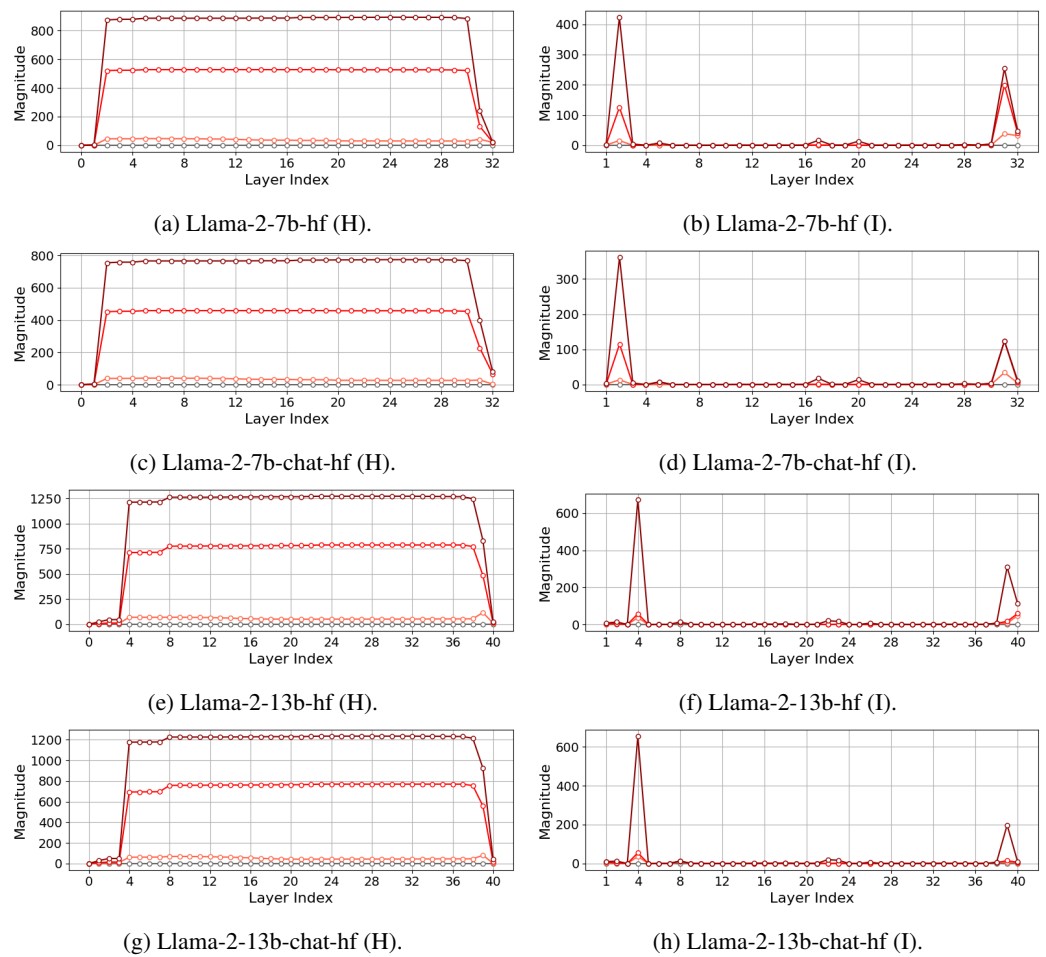

(a) Llama-2-7b-hf (H).

(b) Llama-2-7b-hf (I).

(c) Llama-2-7b-chat-hf (H).

(d) Llama-2-7b-chat-hf (I).

(e) Llama-2-13b-hf (H).

(f) Llama-2-13b-hf (I).

(g) Llama-2-13b-chat-hf (H).

(h) Llama-2-13b-chat-hf (I).

*Figure 19.* (Left) Hidden state and (Right) intermediate state of Llama-2 family.

## D.2. Llama-3 family

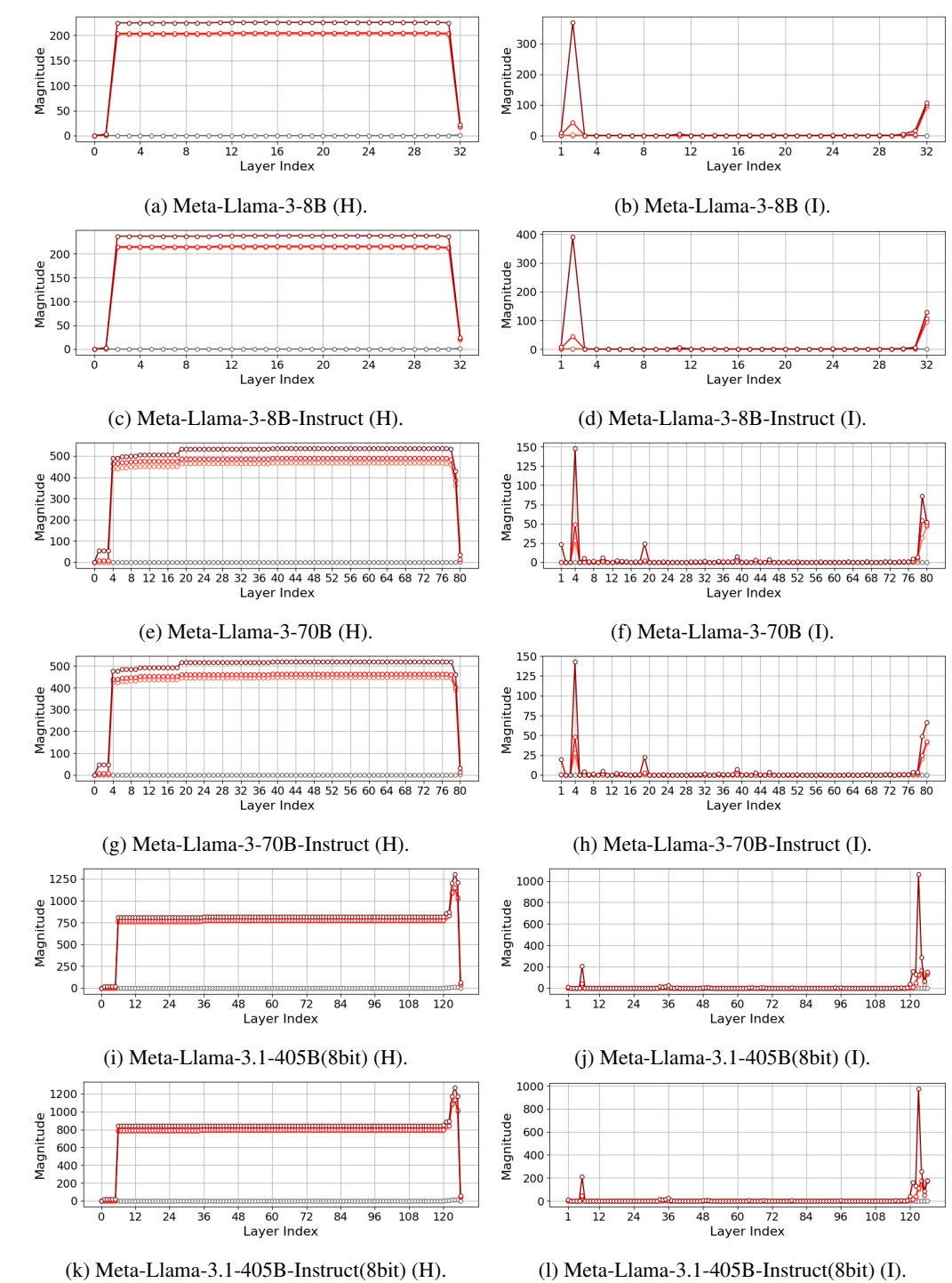

(a) Meta-Llama-3-8B (H).

(b) Meta-Llama-3-8B (I).

(c) Meta-Llama-3-8B-Instruct (H).

(d) Meta-Llama-3-8B-Instruct (I).

(e) Meta-Llama-3-70B (H).

(f) Meta-Llama-3-70B (I).

(g) Meta-Llama-3-70B-Instruct (H).

(h) Meta-Llama-3-70B-Instruct (I).

(i) Meta-Llama-3.1-405B(8bit) (H).

(j) Meta-Llama-3.1-405B(8bit) (I).

(k) Meta-Llama-3.1-405B-Instruct(8bit) (H).

(l) Meta-Llama-3.1-405B-Instruct(8bit) (I).

*Figure 20.* (Left) Hidden state and (Right) intermediate state of Llama-3/3.1 family.

## D.3. Mistral and Mixtral family

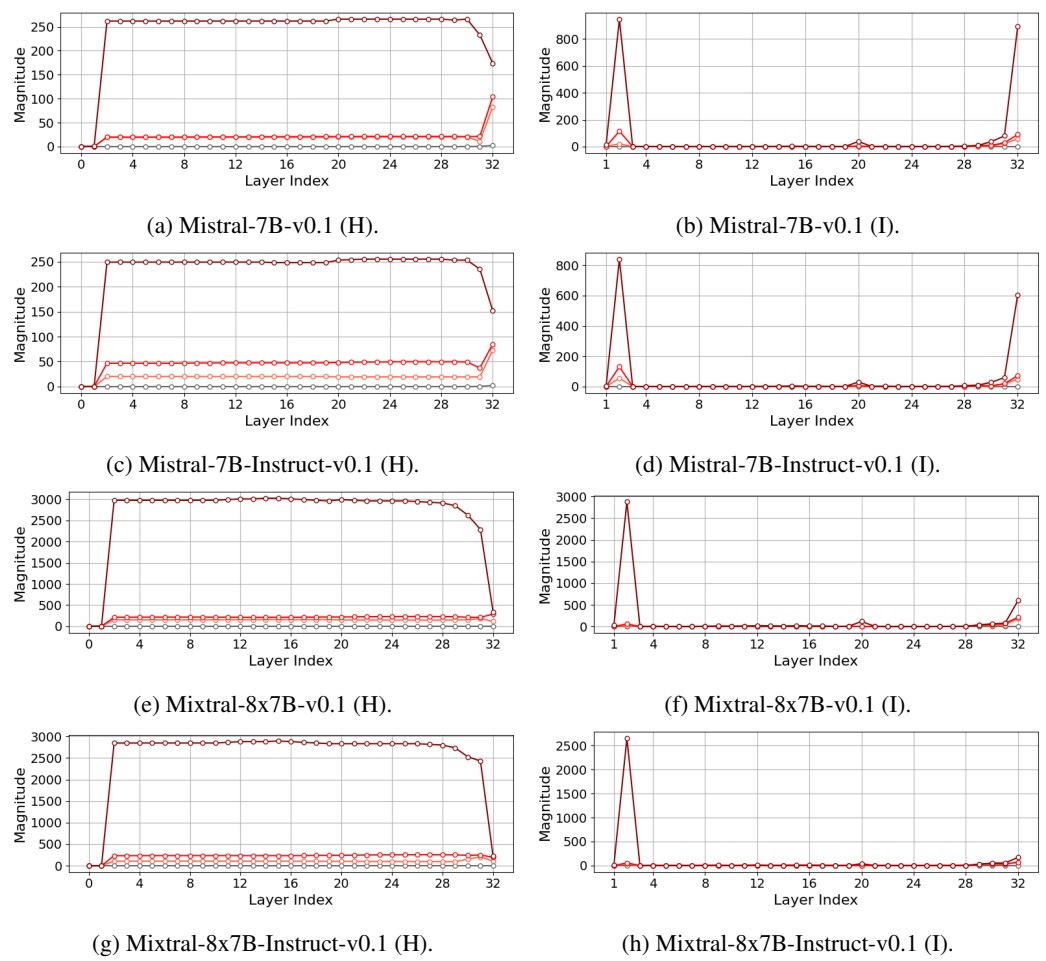

(a) Mistral-7B-v0.1 (H).

(b) Mistral-7B-v0.1 (I).

(c) Mistral-7B-Instruct-v0.1 (H).

(d) Mistral-7B-Instruct-v0.1 (I).

(e) Mixtral-8x7B-v0.1 (H).

(f) Mixtral-8x7B-v0.1 (I).

(g) Mixtral-8x7B-Instruct-v0.1 (H).

(h) Mixtral-8x7B-Instruct-v0.1 (I).

*Figure 21.* (Left) Hidden state and (Right) intermediate state of Mistral and Mixtral family.

## D.4. Phi-3 family

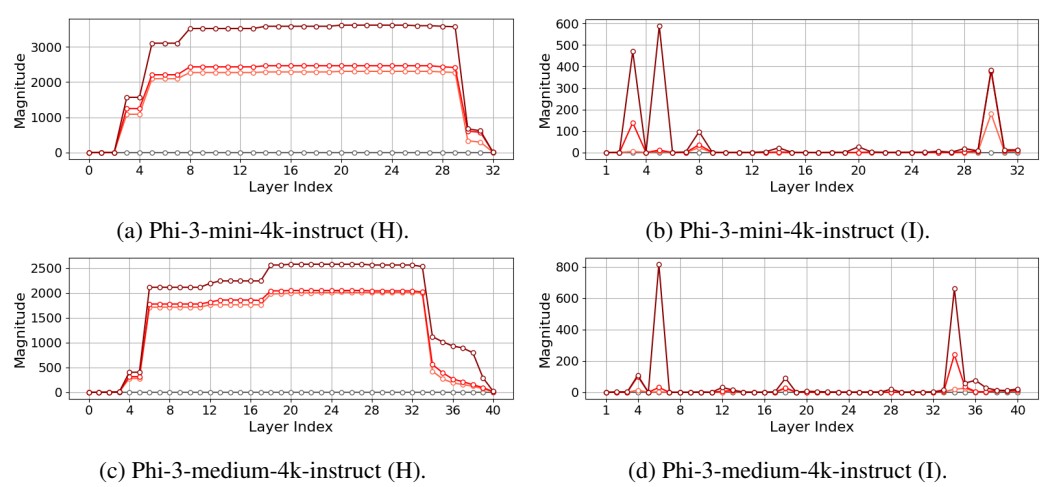

(a) Phi-3-mini-4k-instruct (H).

(b) Phi-3-mini-4k-instruct (I).

(c) Phi-3-medium-4k-instruct (H).

(d) Phi-3-medium-4k-instruct (I).

*Figure 22.* (Left) Hidden state and (Right) intermediate state of Phi-3 family.

## D.5. Gemma-2 family

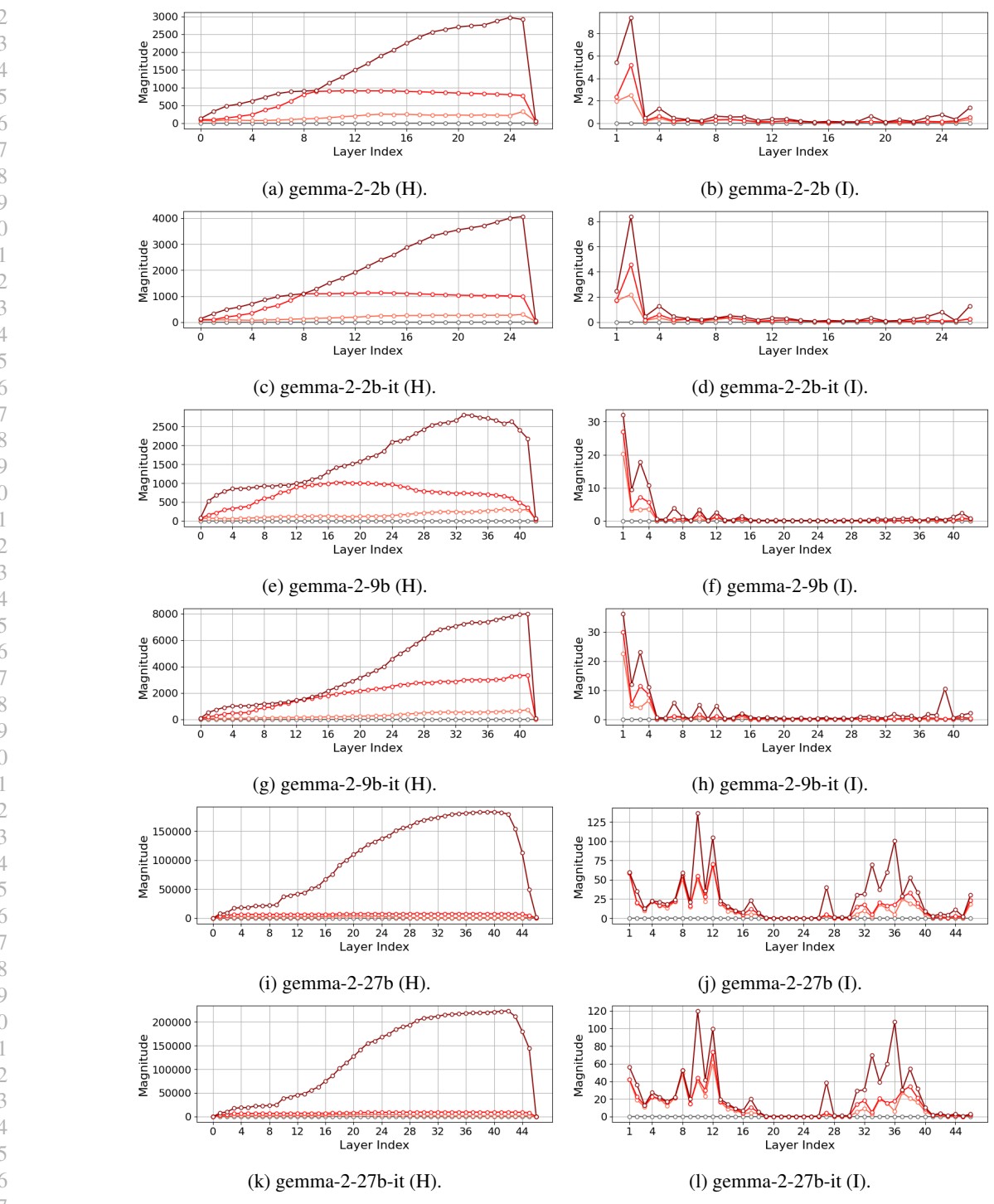

(a) gemma-2-2b (H).

(b) gemma-2-2b (I).

(c) gemma-2-2b-it (H).

(d) gemma-2-2b-it (I).

(e) gemma-2-9b (H).

(f) gemma-2-9b (I).

(g) gemma-2-9b-it (H).

(h) gemma-2-9b-it (I).

(i) gemma-2-27b (H).

(j) gemma-2-27b (I).

(k) gemma-2-27b-it (H).

(l) gemma-2-27b-it (I).

*Figure 23.* (Left) Hidden state and (Right) intermediate state of Gemma-2 family.

# E. Attention Sinks

Figure 24 describes attention after Softmax in the early layers (from layer 1 to layer 8) across various models. Attention sinks are observed in the layers after the massive weights layer. In Llama-2-7B (Figure 24(a)), Mistral-7B (Figure 24(c)), and Mixtral-8x7B (Figure 24(d)), sink tokens are the initial token ('Summer') and the first delimiter token ('.'), discovered by Sun et al. (2024a). In Llama-3-8B (Figure 24(b)) and Phi-3-mini (Figure 24(e)), sink token is the only the initial token ('Summer'). Interestingly, in these five models, it is commonly observed that significant attention is concentrated on non-semantic tokens ('.') before attention sinks occur. However, in Gemma-2 (Figure 24(f)), attention sinks do not happen and attention is primarily assigned to local tokens. Note that what we provide is the average of the heads, and there might be heads that do not fully sink when viewed individually.

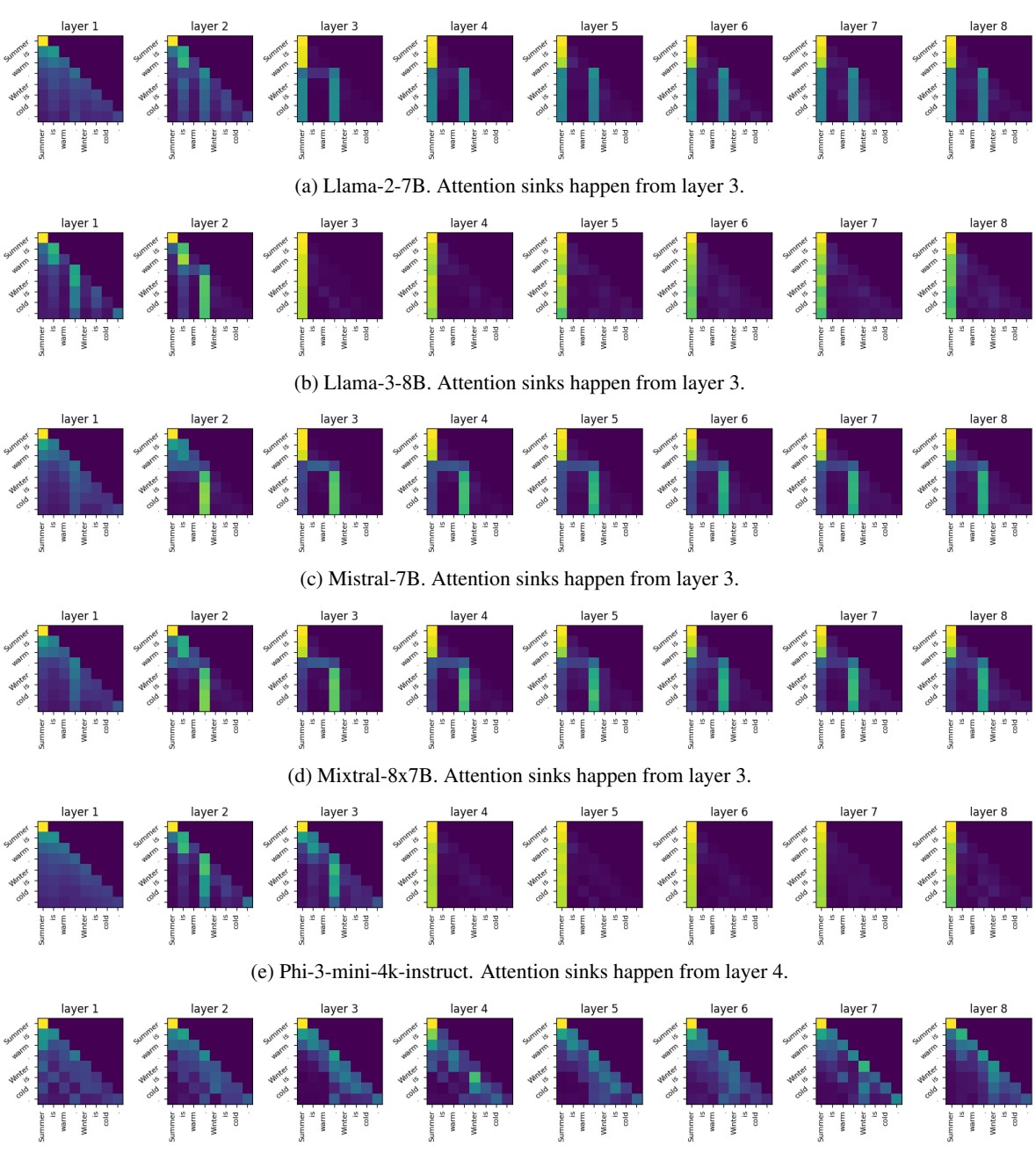

(a) Llama-2-7B. Attention sinks happen from layer 3.

(b) Llama-3-8B. Attention sinks happen from layer 3.

(c) Mistral-7B. Attention sinks happen from layer 3.

(d) Mixtral-8x7B. Attention sinks happen from layer 3.

(e) Phi-3-mini-4k-instruct. Attention sinks happen from layer 4.

(f) gemma-2-2b. Attention sinks do not happen.

*Figure 24.* Attention after Softmax.

# F. Zero-shot Downstream Task across Various LLMs

We address the potential limitations of algorithms based on massive phenomena. Table 8 presents the results on zero-shot downstream tasks across different LLMs. The results show that `MacDrop` is not effective for LLMs that are not sensitive to massive weights, such as Phi-3-medium and Gemma-2 family, as shown in Section 2.

Nevertheless, we believe that clearly demonstrating such limitations is itself meaningful. Most previous and ongoing research addressing massive phenomena have stated that various models exhibit massive phenomena, and have proposed algorithms and experimental results excluding Gemma-2 family. Because we conduct experiments on a wide variety of recent LLMs, we are able to reveal such limitations. Additionally, we would like to emphasize that, similar to previous studies, `MacDrop` demonstrates sufficient feasibility for LLMs exhibiting massive phenomena.

*Table 8.* Zero-shot downstream tasks performance across different LLMs.

| Model | Method | ARC-Easy | ARC-Challenge | BoolQ | PIQA | WinoGrande | Avg. |
|---|---|---|---|---|---|---|---|
| Llama-2-13B | LoRA | 77.1 | 51.9 | 82.2 | 81.6 | 72.8 | 73.1 |
| | + MacDrop | 78.8 | 52.6 | 82.2 | 71.4 | 71.9 | **73.4** |
| | DoRA | 77.4 | 51.8 | 82.0 | 81.6 | 72.8 | 73.1 |
| | + MacDrop | 77.8 | 52.5 | 81.9 | 81.8 | 72.3 | **73.3** |
| Llama-3-8B | LoRA | 79.6 | 58.2 | 83.9 | 82.4 | 75.9 | 76.0 |
| | + MacDrop | 82.9 | 58.3 | 83.9 | 82.6 | 75.0 | **76.5** |
| | DoRA | 80.8 | 57.7 | 83.9 | 82.5 | 75.8 | 76.1 |
| | + MacDrop | 81.9 | 58.2 | 83.9 | 82.2 | 75.6 | **76.4** |
| Llama-3-70B | LoRA | 87.5 | 66.3 | 86.1 | 84.9 | 80.4 | 81.0 |
| | + MacDrop | 87.5 | 66.5 | 86.1 | 85.0 | 80.8 | **81.2** |
| | DoRA | 87.5 | 66.6 | 86.1 | 85.0 | 80.8 | **81.2** |
| | + MacDrop | 87.4 | 66.6 | 86.0 | 85.1 | 80.9 | **81.2** |
| Mistral-7B | LoRA | 78.5 | 54.9 | 84.9 | 82.9 | 75.3 | 75.3 |
| | + MacDrop | 80.9 | 56.7 | 85.0 | 83.0 | 75.3 | **76.2** |
| | DoRA | 78.4 | 55.1 | 85.0 | 82.9 | 75.1 | 75.3 |
| | + MacDrop | 80.6 | 56.7 | 85.3 | 82.9 | 75.1 | **76.1** |
| Phi-3-mini | LoRA | 72.5 | 53.7 | 86.4 | 80.1 | 74.0 | 73.3 |
| | + MacDrop | 75.0 | 54.7 | 86.2 | 80.5 | 74.0 | **74.1** |
| | DoRA | 72.3 | 53.3 | 86.4 | 80.0 | 73.6 | 73.1 |
| | + MacDrop | 72.9 | 53.5 | 86.3 | 80.0 | 73.7 | **73.3** |
| Phi-3-medium | LoRA | 81.4 | 62.2 | 88.7 | 82.6 | 76.4 | **78.3** |
| | + MacDrop | 81.2 | 61.9 | 88.7 | 82.5 | 76.4 | 78.1 |
| | DoRA | 80.92 | 62.0 | 88.6 | 82.4 | 76.2 | **78.0** |
| | + MacDrop | 80.8 | 61.9 | 88.5 | 82.4 | 76.2 | 77.9 |
| Gemma-2-2b | LoRA | 81.6 | 54.4 | 79.8 | 79.2 | 68.7 | **72.7** |
| | + MacDrop | 81.5 | 54.2 | 79.6 | 79.2 | 68.6 | 72.6 |
| | DoRA | 81.4 | 54.0 | 79.6 | 79.3 | 68.7 | **72.6** |
| | + MacDrop | 81.6 | 53.9 | 79.4 | 79.3 | 68.7 | **72.6** |
| Gemma-2-9b | LoRA | 89.6 | 69.0 | 86.5 | 82.8 | 75.3 | **80.6** |
| | + MacDrop | 89.2 | 68.4 | 86.5 | 82.8 | 75.2 | 80.4 |
| | DoRA | 89.4 | 68.9 | 86.2 | 82.7 | 75.8 | **80.6** |
| | + MacDrop | 89.2 | 68.5 | 86.3 | 82.8 | 75.7 | 80.5 |
| Gemma-2-27b | LoRA | 87.5 | 69.0 | 86.0 | 84.1 | 80.6 | **81.4** |
| | + MacDrop | 87.3 | 68.8 | 86.0 | 84.2 | 80.5 | **81.4** |
| | DoRA | 88.5 | 69.4 | 85.6 | 84.5 | 80.0 | **81.6** |
| | + MacDrop | 88.1 | 69.2 | 85.7 | 84.3 | 80.0 | 81.5 |

# G. Generation Tasks

We evaluate on the generated texts of the same models in Section 4.1 using the Spec-Bench dataset (Xia et al., 2024). This benchmark includes six subtasks, each containing 80 instances: multi-turn (MT) conversation from MT-bench (Zheng et al., 2023), translation from WMT14 DE-EN (Bojar et al., 2014), summarization from CNN/Daily Mail (Nallapati et al., 2016), question answering (QA) from Natural Questions (Kwiatkowski et al., 2019), mathematical reasoning from GSM8K (Cobbe et al., 2021), and retrieval-augmented generation (RAG) from Natural Questions (Kwiatkowski et al., 2019). We utilize the direct assessment of Prometheus-2-7B (Kim et al., 2024) to evaluate the generated texts using a 5-point Likert scale. Prometheus-2-7B is an open-source language model specifically designed for evaluation purposes[6]. Table 9 presents the results on generation tasks. MT-1 and MT-2 indicate the first turn and second turn, respectively. Unfortunately, `MacDrop` shows limited performance improvements in generation tasks. Table 10 provides examples of the generated texts and judgements. Assistant A and B indicate Llama-3-8B with and without `MacDrop`, respectively.

*Table 9.* Generation tasks performance measured by Prometheus-2-7B.

| Model | Method | MT-1 | MT-2 | translation | summarization | QA | math reasoning | RAG | Avg. |
|---|---|---|---|---|---|---|---|---|---|
| Llama-3-8B | LoRA | 3.71 | 3.54 | 4.50 | 3.24 | 4.35 | 3.64 | 3.74 | **3.82** |
| | + `MacDrop` | 3.76 | 3.49 | 4.51 | 3.39 | 4.29 | 3.50 | 3.71 | 3.81 |
| | DoRA | 3.85 | 3.71 | 4.56 | 3.34 | 4.35 | 3.59 | 3.73 | 3.80 |
| | + `MacDrop` | 3.79 | 3.41 | 4.55 | 3.26 | 4.29 | 3.79 | 3.85 | **3.85** |
| Mistral-7B | LoRA | 3.55 | 3.30 | 4.61 | 3.33 | 4.41 | 3.45 | 4.05 | **3.81** |
| | + `MacDrop` | 3.75 | 3.42 | 4.58 | 3.26 | 4.35 | 3.30 | 3.98 | **3.81** |
| | DoRA | 3.55 | 3.29 | 4.64 | 3.34 | 4.49 | 3.35 | 3.88 | 3.79 |
| | + `MacDrop` | 3.49 | 3.29 | 4.59 | 3.52 | 4.41 | 3.33 | 3.95 | **3.80** |

*Table 10.* Example of generation task (translation).

| | |
|---|---|
| User | Translate German to English: St. Hubertus war der Sage nach ein recht rücksichtsloser Jäger, der durch eine Vision bekehrt und später zum Bischof von Lüttich ernannt wurde. |
| Reference answer | St. Hubert was, according to legend, a truly ruthless hunter, who converted following a vision and was later appointed Bishop of Lüttich. |
| Assistant A | According to legend, St. Hubertus was a rather ruthless hunter who was converted and later appointed as the Bishop of Liège after a vision. |
| Judgment for A | The response provided a translation that was generally accurate and relevant to the user's request. However, there were minor inaccuracies in the phrasing, such as "rather ruthless" instead of "truly ruthless," which slightly deviated from the original text. Additionally, the term "Liège" was used instead of "Lüttich," which is the correct name of the city in the original German text. Despite these minor issues, the response still conveyed the main points of the legend about St. Hubertus. Therefore, while the response was helpful and mostly aligned with the user's inquiry, it did not perfectly cater to the user's needs due to the slight inaccuracies in the translation. So the overall score is 4. |
| Assistant B | According to legend, St. Hubertus was a rather reckless hunter who was converted and later appointed as the Bishop of Liège after a vision. |
| Judgment for B | The response provided a translation that was generally accurate and relevant to the user's request. However, there were slight deviations from the original text. For instance, the term "rather reckless" is less precise than "truly ruthless," which could lead to some ambiguity. Additionally, the term "Liège" is used instead of "Lüttich," which is a minor inaccuracy. Despite these minor issues, the response still managed to convey the main points of the original text, making it useful for the user's needs. Therefore, the response aligns well with the user's inquiry, with only rare inaccuracies, and thus meets the criteria for a score of 4. |

---

[6]https://github.com/prometheus-eval/prometheus-eval

# H. Related work

The attention sinks phenomenon and their importance, uncovered by Xiao et al. (2024), have been widely used to compress key-value caches. For quantization, KVQuant (Hooper et al., 2024) applies attention sink-aware quantization, which retains only the first token in fp16. CushionCache (Son et al., 2024) inserts sink tokens into the prefix to mitigate massive activations in the middle of the sequence, enhancing the performance of quantized models. For token eviction and token merging, sink tokens are never evicted or merged; they remain unchanged (Xiao et al., 2024; Ge et al., 2024; Li et al., 2024; Zhang et al., 2023; Wang et al., 2024; Zhang et al., 2024).

Nevertheless, there has been limited in-depth research on the phenomenon itself. In fact, the idea of global attentions, such as [CLS] and [SEP] tokens–similar to attention sinks–was introduced and emphasized even before the LLM era (Zaheer et al., 2020; Beltagy et al., 2020). In the LLM era, Yu et al. (2024b) showed that sink tokens can appear not only at the beginning of a sentence but also in the middle, and they are often shown to be nonsemantic (e.g., '.'). Sun et al. (2024a) discovered the presence of massive activations in the hidden state space of sink tokens, demonstrating that massive activations trigger the attention sinks phenomenon. Meanwhile, in vision transformers, similar phenomenon is observed (Darcet et al., 2024). They showed that training with register tokens, which is additional meaningless tokens similar to sink tokens, resulted in improved dense prediction and interpretability. Different from previous work, we explore this phenomenon in the weight space.

Specifically, we define the massive weights in an activation-aware manner using the *bos* token. Similarly, Wanda (Sun et al., 2024b) with a structured pruning (An et al., 2024) and AWQ (Lin et al., 2024) calculate weight importance scores based on a small of calibration data. However, it is important to note that the massive weights are confined to a specific single layer, whereas Wanda and AWQ identify important weights within every linear layer. In other words, the massive weights would be included among those selected through Wanda or AWQ. Our contribution focuses more deeply on a narrowly defined aspect compared to these studies.

Concurrently with our work, Yu et al. (2024a) have defined "super weight," which is a single weight that can significantly influence the performance of LLMs. To compare more specifically, we identify that the intermediate state in the layer $l$ possesses massive activations. Accordingly, we define massive weight as the corresponding rows of $W_{up}$. However, Yu et al. (2024a) take into account both the intermediate state and the subsequent hidden state. In other words, they define super weight as the intersection of the corresponding column of $W_{down}$ to the super activation in the intermediate state and the corresponding row of $W_{down}$ to the super activation in the hidden state, at the layer $l$. In summary, we examine the intermediate state exclusively from an output perspective (i.e., $W_{up}$), while their approach appears to have considered it from an input perspective (i.e., $W_{down}$), incorporating the hidden state.