# OpenReview forum: "House of Cards: Massive Weights in LLMs"
_ICML.cc/2025/Conference — Submitted to ICML 2025_

### Official Review · Reviewer_jmDs · 2025-02-28

**Overall Recommendation:** 3

**Summary:**

This paper identifies the phenomenon of "massive weights", which contributes to a know phenomenon called "massive activations". These massive weights only take up a small part of LLMs, but significant impact the model performance. When zeroing the top-k massive weights, LLMs experience serious performance degradation. This is called top-k zeroing attack in this paper.

To reduce the reliance on massive weights, the authors propose a massive weights curriculum dropout (MacDrop) during the LoRA or DoRA fine-tuning. The results show that with MacDrop, the performance of fine-tuned LLMs is better. And the LLMs are more resilient to top-k zeroing attack.

## update after rebuttal

I think the authors alleviate my concerns to some extent. I increased my rating to 3 as this paper identifies the phenomenon of "massive weights", which contributes to the mechanical interpretability of LLMs. But I think the proposed MacDrop and top-k zero attacking are less practical. I assume these two approaches could be regarded as manners to explain the phenomenon. This prevents me from rating higher.

**Claims And Evidence:**

I think the major claims 1) massive activations are related to massive weights, which are small in quantity but significantly impact the performance; 2) proposed MacDrop could reduce the reliance of LLMs on massive weights are justified through the visualisations and empirical studies.

**Essential References Not Discussed:**

I think the essential references are discussed in this paper.

**Experimental Designs Or Analyses:**

Yes, I think experimental designs and analyses make sense.

**Methods And Evaluation Criteria:**

I think proposed methods and evaluation criteria make sense.

**Other Comments Or Suggestions:**

I think the key message here is that massive weights, which are related to massive activations, can significantly impact the model performance. StreamingLLM [1] has already shown that without the initial tokens, the PPLs will soar. This hints that attention sink is essential to model performance. Back to the massive weights, I conjecture that when zeroing the massive weights, massive activations and attention sink disappear, and then model performance drops.


[1] Xiao et al. Efficient Streaming Language Models with Attention Sinks. ICLR 2024.

**Other Strengths And Weaknesses:**

The motivations behind the proposal of MacDrop is "massive weights are predominantly learned during pre-training, and that zeroing them can severely undermine LLMs". However, why we need to relax the reliance of LLMs on massive weights?

I am aware that this paper showcases the top-k zeroing attack. However, in the real applications, will the top-k zeroing attack be used? For production LLMs, which one cannot access the weights, such attack cannot be conducted. For open-sourced LLMs, I think it should be easy to make LLM performance drop by randomly editing several model parameters. Therefore, the motivation of MacDrop should be further justified.

**Questions For Authors:**

Why do Gemma 2 models have no attention sink? Do you have intuitions behind this?

In Appendix C.5, I find that only without BOS, there is no attention sink phenomenon. Have you checked whether Gemma models can still have normal performance without BOS? According to [1], when fine-tuning Gemma models, BOS token is required to be added. Otherwise, the loss is much higher. Therefore, if without BOS, model performance becomes much worse. It is not necessary to discuss whether there is attention sink.

Although it is not related to my ratings, just from curiosity, how about the massive weights for LLMs which have no massive activations? In [2], they showed that using learnable kv bias could mitigate massive activations. In such cases, will massive weights also disappear?

[1] https://unsloth.ai/blog/gemma-bugs \
[2] Sun et al. Massive Activations in Large Language Models. COLM 2024.

**Relation To Broader Scientific Literature:**

The authors attribute the massive activations to massive weights. The conclusions in this paper may be interesting to related literature about massive activations, or downstream applications, such as model quantization.

The proposed MacDrop seems to improve the performance of LoRA/DoRA to some extend. I think it may be interested to communities about general fine-tuning topics.

**Theoretical Claims:**

There are no theoretical claims in this paper.

---

> ### Author Rebuttal · Authors · 2025-03-27
>
> Thank you for your insightful review. We would like to answer the concerns raised in **Weaknesses**, **Comments Or Suggestions**, and **Questions**.
>
> ---
> ### Weaknesses
>
> - First, because our paper deals with open-source LLMs that allow access to weights, we begin by addressing random weights editing.
>   - As you mentioned, random weights editing is a simpler attack. However, although it may show whether the model's performance decreases (or does not decrease), it has limitations in revealing whether the model's performance depends on particular weights.
>   - Therefore, by showing that the zeroing attack on **targeted specific weights** leads to severe degradation, we highlighted the existence and significance of massive weights.
>
> - We also recognize that such attacks are not feasible on production LLMs.
>   - **We used such attacks as a means to analyze and gain a deeper understanding** of the weights in LLMs, rather than focusing on the attack itself.
>   - In this regard, we have clarified this point in **Impact Statement**
>
> - Finally, we acknolwedge that instead of mitigating massive weights, it is possible to utilize them, as demonstrated by other algorithms introduced in Related Work (Appendix H).
>   - However, from the perspective of a model developer, who can access the model weights, it is reasonable to think that one does not want a large-scale model, developed at significant cost, to exhibit over-dependence on just a few weights.
>   - For this reason, we would like to highlight the motivation for mitigating massive weights rather than utilizing them, because developing more robust models is necessary. Accordingly, we have expressed this point in **Impact Statement**
>
> ---
>
> ### Comments Or Suggestions
> - Thank you for your suggestions. For verification, we examine the intermediate and hidden states across layers (like Figure 3) and the attention maps (like Figure 2) using the top-5 massive weights zeroed-out models listed in Table 1.
>
> - As expected, **the massive activations in the states disappear, and the attention sink phenomenon does not occur**.
>
> - We kindly ask for your understanding, as we are currently unable to update the manuscript and can only provide this information in text form. Figures will be updated.
>
> ---
>
> ### Questions
> - We argue that the lower sensitivity of the Gemma-2 family can be attributed to the LayerNorm layer, discussed with **Equation 3**.
>   - Very recently, a related paper [1] has provided an analysis indicating that the architectural design shown in Equation 3—referred to as Peri-LN—preserves **gradient stability** and prevents the emergence of massive activations observed in Pre-LN (e.g., in the Llama family).
>   - In detail, Pre-LN applies LayerNorm only to the input, while allowing the output to pass through the residual path. As a result, massive activations generated may accumulate. Peri-LN applies normalization at both the input and output of each sub-layer. This two normalization prevents gradient explosion.
>   - We consider this paper to offer deeper insights into the functional role of LayerNorm, which we will cover in an updated manuscript.
>
> - During rebuttal, we checked zero-shot performance of the Gemma-2 family depending on the bos token during both PEFT and evaluation. **Our default setting use the bos token for both PEFT and evaluation**. The averaged zero-shot performance is as follows:
> | PEFT/Evaluation | O/O | X/O | O/X | X/X |
> |---|---|---|---|---|
> | Gemma-2-2B + LoRA | **72.7** | 71.9 | 44.3 | 63.4 |
> | Gemma-2-9B + LoRA | **80.6** | 79.7 | 46.1 | 73.8 |
>   - This results indicate that not using the bos token during evaluation (i.e., O/X case) leads to a greater performance drop compared to not using it during PEFT (i.e., X/O case).
>   - We found information messages related to this issue in lm-eval github [2]. And, we modify below line for not using the bos token.
> ```
> if "gemma" in getattr(self.config, "model_type", ""):
>     self.add_bos_token = True
>     eval_logger.info(
>         f"Model type is '{self.config.model_type}', part of the Gemma family--a BOS token will be used as Gemma underperforms without it."
>     )
> ```
>
> - When considering the Gemma-2 family, we repeatedly encountered the same concerns. In our opinion, since massive weights are defined in relation to massive activations, it is appropriate to consider that massive weights also disappear.
>   - In other words, if the KV bias in [3] is properly learned, we expect that the model’s performance will be preserved even if the weights—previously defined as massive weights—are attacked.
>
> ---
>
> [1] Peri-LN: Revisiting Layer Normalization in the Transformer Architecture
> [2] https://github.com/EleutherAI/lm-evaluation-harness/blob/main/lm_eval/models/huggingface.py
> [3] Massive Activations in Large Language Models

---

> > ### Comment · Reviewer_jmDs · 2025-04-02
> >
> > Why do you think that Gemma-2 family has lower sensitivity due to the LayerNorm layer? I have not checked the structure of Gemma-2 models. Do you think Gemma-2 adopt a Peri-LN structure as you used this example?

---

> > > ### Author Response · Authors · 2025-04-03
> > >
> > > Thank you for your rapid response.
> > >
> > > ---
> > >
> > > We can easily check the structure of Llama models from transformers github [1]. Please refer to \_\_init\_\_ and forward methods of class LlamaDecoderLayer.
> > > - In a single layer, **two LayerNorm** layers exist, which is described as **Equation (1)** in our paper. In addition, this architecture is called as pre-LN in [3].
> > >  - From the perspective of gradient stability, this structure is analyzed to have potential for gradient explosion [3].
> > > ```
> > > self.input_layernorm = LlamaRMSNorm(config.hidden_size, eps=config.rms_norm_eps)
> > > self.post_attention_layernorm = LlamaRMSNorm(config.hidden_size, eps=config.rms_norm_eps)
> > > ```
> > >
> > >
> > > Similarly, we can check the structure of Gemma-2 models from transformers github [2]. Please refer to \_\_init\_\_ and forward methods of class Gemma2DecoderLayer.
> > > - In a single layer, **four LayerNorm** layers exist, which is described as **Equation (3)** in our paper. In addition, this architecture is called as peri-LN in [3].
> > > - It is showed that Peri-LN can mitigate and gradient instability [3].
> > > - In addition, Gemma-2 is introduced as one of representative models using peri-LN in [3].
> > > ```
> > > self.input_layernorm = Gemma2RMSNorm(config.hidden_size, eps=config.rms_norm_eps)
> > > self.post_attention_layernorm = Gemma2RMSNorm(config.hidden_size, eps=config.rms_norm_eps)
> > >
> > > self.pre_feedforward_layernorm = Gemma2RMSNorm(config.hidden_size, eps=config.rms_norm_eps)
> > > self.post_feedforward_layernorm = Gemma2RMSNorm(config.hidden_size, eps=config.rms_norm_eps)
> > > ```
> > >
> > > In summary, the evidence clearly supports the conclusion that Gemma-2 incorporates the peri-LN architecture in its design.
> > >
> > > ---
> > > [1] https://github.com/huggingface/transformers/blob/main/src/transformers/models/llama/modeling_llama.py
> > > [2] https://github.com/huggingface/transformers/blob/main/src/transformers/models/gemma2/modeling_gemma2.py
> > > [3] Peri-LN: Revisiting Layer Normalization in the Transformer Architecture

---

### Official Review · Reviewer_yuFo · 2025-03-11

**Overall Recommendation:** 3

**Summary:**

The paper studies and localizes the weight vector that causes the massive activation, and observes that the massive weights exist in different model families and even MoE models. The paper also found that setting the the massive weights (of one layer) to zero will complete destroy the model while setting the complement weights only hurts mthe odel by small portion.

Based on the findings that massive weights are important, the authors propose the MacDrop approach to drop the massive weights during the parameter-efficient fine-tuning to penalize the impact of those massive weights.

**Update after rebuttal**: My latest reply reflected my final update.

**Claims And Evidence:**

No problematic claims.

**Essential References Not Discussed:**

N/A

**Experimental Designs Or Analyses:**

I checked all the experimental results.

**Methods And Evaluation Criteria:**

The methods and evaluation make sense.

**Other Comments Or Suggestions:**

In the abstract, it would be great to mention the massive weights are defined in just one layer. Or it will sound like the majority of weights of the whole models are zeroed out when reading lines 28-29 "However, when all weights except for massive weights are set to zero".

**Other Strengths And Weaknesses:**

[Strength]

* The paper explores the massive weights of various model families.
* The results on retaining top-k massive weights is better than dropping top-k massive weights are interesting.

[Weakness]

Major:
* The proposed method, MacDrop, does not show a clear improvement over not using it (Tables 2 and 3) when considering results from a single run. It would be more compelling if the results were averaged over multiple runs to ensure robustness.
* The novelty of zeroing out massive weights and its impact on performance appears limited, as it is conceptually similar to zeroing out massive attention, which has been explored in [1].
* Given the previous two limitations, the significance of defining massive weights remains unclear. It would be helpful to demonstrate more applications to highlight their value.

[1] Massive Activations in Large Language Models

**Questions For Authors:**

N/A

**Relation To Broader Scientific Literature:**

The findings in this paper might motivate people to pre-train/post-training models to mitigate the superweights.

**Theoretical Claims:**

There are no theoretical claims.

---

> ### Author Rebuttal · Authors · 2025-03-26
>
> Thank you for your clear review. We will revise the abstract to make it crystal clear, reflecting your comments accordingly.
>
> During the rebuttal process, we focus on addressing the weaknesses in **Weaknesses**.
>
> ---
> ### Weaknesses
> - (1) Multiple runs
>   - To address this, we conduct **3 runs** with different seeds. The results consistently show that MacDrop outperforms the baseline (i.e., without MacDrop) in all cases. Namely, the overall trends are consistent with the single-run results reported in the original tables (**Tables 2 and 3**).
>   - Zero-shot downstream tasks (average over 3 runs)
> Model | Method          | Avg. ± Std |
> |--------|-------------------|-----------------------------|
> | Llama-3-8B | LoRA             | 75.97 ± 0.08          |
> | Llama-3-8B | LoRA + MacDrop   | **76.14** ± 0.28      |
> | Llama-3-8B | DoRA             | 76.03 ± 0.11          |
> | Llama-3-8B | DoRA + MacDrop   | **76.37** ± 0.07      |
> | Mistral-7B | LoRA             | 75.19 ± 0.09          |
> | Mistral-7B | LoRA + MacDrop   | **76.33** ± 0.10     |
> | Mistral-7B | DoRA             | 75.32 ± 0.03         |
> | Mistral-7B | DoRA + MacDrop   | **76.19** ± 0.06      |
>   - Long context tasks (average over 3 runs)
> Model | Method          | Avg. ± Std |
> |--------|-------------------|-----------------------------|
> | Llama-3-8B | LoRA             | 39.09 ± 0.08          |
> | Llama-3-8B | LoRA + MacDrop   | **40.85** ± 0.24      |
> | Llama-3-8B | DoRA             | 38.87 ± 0.27          |
> | Llama-3-8B | DoRA + MacDrop   | **39.68** ± 0.17      |
> | Mistral-7B | LoRA             | 37.10 ± 0.15          |
> | Mistral-7B | LoRA + MacDrop   | **38.04** ± 0.20     |
> | Mistral-7B | DoRA             | 37.00 ± 0.19         |
> | Mistral-7B | DoRA + MacDrop   | **37.06** ± 0.27      |
>
> - (2) Novelty of zeroing attack
>   - We did not intend to emphasize the novelty of the zeroing attack. Rather, **we already clarified the similarities and differences** with the attack used in [1] in **Section 2.3**.
>     - "In essence, this attack is very similar to the one proposed in [1], where massive activations in the hidden state are zeroed out in a single layer. The difference is that their attack targets the hidden state, while our attack targets the intermediate state."
>   - In other words, we would like to highlight that while the method of attack is the same, the target is different. Our intention is to emphasize that the massive phenomenon appears **earlier than a hidden state (i.e., intermediate state)** identified in [1].
>
> - (3) Significance of defining massive weights
>   - We disagree with the claim that the significance of the massive weights remains unclear. Through **Table 1**, we demonstrated that zeroing out the massive weights (approximately 0.0005% of the total weights) leads to a complete degradation in performance. This alone suggests that these weights are indeed significant.
>   - Furthermore, our responses to weaknesses (1) and (2) provide further clarification on this matter.
>   - In addition, **based on massive weights**, we proposed MacDrop and demonstrated its effectiveness in terms of both performance (**Tables 2 and 3**) and robustness (**Table 4**). Namely, MacDrop is one of the applicable algorithms that consider massive weights. Its effectiveness underscores the importance of considering massive weights in the design for parameter-efficient fine-tuning.
>
> ---
>
> [1] Massive Activations in Large Language Models

---

> > ### Comment · Reviewer_yuFo · 2025-04-05
> >
> > Thanks to the authors for the rebuttal contents. Although I still hold small concerns about the applications of massive weights other than MacDrop, I have increased my rating accordingly.

---

> > > ### Author Response · Authors · 2025-04-07
> > >
> > > Thank you for your thoughtful response.
> > >
> > > As you mentioned, we believe the observation on massive weights holds broader potential for future research. That said, we would like to gently emphasize that MacDrop is the first algorithm explicitly built upon the concept of massive weights.

---

### Official Review · Reviewer_Teii · 2025-03-15

**Overall Recommendation:** 3

**Summary:**

This work investigates the massive weights phenomenon in large language models (LLMs). The authors observe that massive weights are strongly associated with the initial BOS token, though the results varies across different models. They also find that massive activations first emerge in the intermediate layers of MLP blocks within the early layers of the model. Furthermore, masking out these massive weights leads to a significant performance drop, highlighting their critical impact.

To address this, the authors propose MacDrop, a parameter-efficient fine-tuning method that employs a curriculum dropout strategy to gradually reduce reliance on massive weights during fine-tuning, thereby improving overall model performance.

### After Rebuttal ###

Thanks for the responses that addressed my concerns. I keep my original score.

**Claims And Evidence:**

The claims are supported by clear and convincing evidence.

**Essential References Not Discussed:**

None.

**Experimental Designs Or Analyses:**

The proposed methods are evaluated with different models and tasks, as well as several ablation studies.

**Methods And Evaluation Criteria:**

the evaluattion make sense

**Other Comments Or Suggestions:**

None

**Other Strengths And Weaknesses:**

- The models are fine-tuned for three epochs—could the authors clarify which epoch’s performance is reported? Since extended fine-tuning can sometimes degrade performance, it is important to specify how checkpoint is selected.

**Questions For Authors:**

Are there any insights whether such massive weights in LLM is a benefit or artifact for LLM's ability?

**Relation To Broader Scientific Literature:**

This work serves as a complementary study to the massive weights phenomenon observed in previous research, providing additional insights into its characteristics and impact on model performance. And the authors use such phenomenon for developing a parameter-efficient finetuning methods.

**Theoretical Claims:**

There are no theoretical claims

---

> ### Author Rebuttal · Authors · 2025-03-26
>
> Thank you for your clear review. We would like to answer the concerns raised in **Weaknesses** and **Questions**.
>
> ---
>
> ### Weaknesses
>
> - We used the trained model for 3 epochs (i.e., the last checkpoint) for all experients. We will update this detail in **Section 4** clearly.
>
> ---
>
> ### Questions
>
> - In our view, massive weights can be regarded as artifacts; however, we believe they also offer advantages. In particular, while they introduce undesirable over-dependence, they contribute positively in areas such as compression.
>
> - First, it is widely recognized that the cost of model pre-training is exceptionally high. Therefore, model developers likely do not want trained models to be totally disrupted by very simple attacks, as shown in **Table 1** of our paper. Based on these results, we believe that the massive weights are not intentionally designed.
>
> - Nevertheless, these massive weights, predominantly learned during pre-training, have a significant impact on the capabilities of LLMs, providing considerable advantages. In essence, they can be interpreted as a means of densely concentrating information. Consequently, we think that LLM algorithms such as streaming generation [1] and quantization [2] have improved based on this underlying principle.
>
> ---
> [1] Efficient Streaming Language Models with Attention Sinks, ICLR 2024
> [2] KVQuant: Towards 10 Million Context Length LLM Inference with KV Cache Quantization, NeurIPS 2024

---

### Official Review · Reviewer_gCmN · 2025-03-17

**Overall Recommendation:** 2

**Summary:**

The paper focuses on understanding the internal mechanisms of Large Language Models (LLMs).

The authors observe that large activations in LLMs, which appear in specific feature dimensions of hidden states, introduce bias by emphasizing the corresponding token.

They identify that these large activations originate from the intermediate state of a feed-forward network module in an early layer.

The authors define “top-k massive weights” as the weights that contribute to the dimensions with the top-k magnitudes in the intermediate state.

They find that setting these massive weights to zero disrupts the functionality of LLMs, while zeroing out all other weights results in a relatively minor performance drop.

Based on this observation, the authors propose a method called MacDrop (massive weights curriculum dropout) to rely less on massive weights during parameter-efficient fine-tuning.

MacDrop applies dropout to the pre-trained massive weights, starting with a high dropout probability and gradually decreasing it as fine-tuning progresses.

The authors demonstrate that MacDrop improves performance and robustness through various experiments.

**Claims And Evidence:**

Claim 1: Massive activations in LLMs introduce bias by overemphasizing specific tokens.

Evidence: The authors provide examples in Figure 1(b) where attacking (zeroing out) the top-5 massive weights in the Llama-3-8B-Instruct model leads to the model repeating the user prompt, indicating a disruption of functionality.

Claim 2: Massive activations originate from the intermediate state of a feed-forward network module in an early layer.

Evidence: The authors trace various states in early layers and observe that the intermediate state inter l within an early layer l exhibits massive activations before they appear in the hidden state hl.  Figure 3 and related descriptions support this claim.

Claim 3: Top-k massive weights (weights contributing to top-k magnitudes in the intermediate state) are crucial for LLM functionality.

Evidence: The authors conduct attacks on Llama-3-8B, Llama-3-70B, and Llama-3.1-405B (8bit) by zeroing out the top-5 massive weights and, conversely, retaining only the top-5 massive weights.  Zeroing out the top-k massive weights severely disrupts LLM performance, while retaining only these weights leads to a relatively minor performance drop.  This demonstrates the significant impact of these weights.
Claim 4: MacDrop (massive weights curriculum dropout) improves performance and robustness in parameter-efficient fine-tuning.

Evidence: The authors conduct experiments on zero-shot downstream tasks and long context tasks, showing that MacDrop generally enhances model performance.  Table 3 and Table 2 provide specific results supporting this claim.  Additionally, Table 4 demonstrates that fine-tuned models with MacDrop exhibit better performance under the top-3 zeroing attack, indicating improved robustness.

Overall, the claims made in the paper are supported by the evidence provided, including experimental results, ablation studies, and comparative analyses across different LLM architectures.

**Essential References Not Discussed:**

Based on the context you provided, here are some related works that are essential to understanding the key contributions of the paper but are not currently cited or discussed:

Weight Pruning Techniques: The paper discusses the importance of identifying crucial weights in LLMs. This is closely related to weight pruning techniques, which aim to remove less important weights from a neural network to reduce its size and computational cost. Some classical works in weight pruning may provide additional context.

Neural Network Interpretability: The paper touches on understanding the internal mechanisms of LLMs, which falls under the broader field of neural network interpretability. Research in this field aims to provide insights into how neural networks make decisions.

Regularization Techniques: The MacDrop method is a regularization technique that applies dropout to specific weights. Other regularization methods, such as L1 or L2 regularization, are commonly used to prevent overfitting in neural networks and could be relevant for comparison.

**Experimental Designs Or Analyses:**

Based on my review, here are some potential issues and questions related to the experimental designs or analyses in the paper:

Ablation Study on Dropout Scope:

The ablation study in Section 4.3.1 examines the effect of dropout scope (all weights, massive weights, all weights except for massive weights) and probability.
The study concludes that applying dropout solely to massive weights can surpass the original performance, but strong dropout on massive weights deteriorates performance.
It would be interesting to see a more detailed analysis of why this occurs. Is it simply a matter of finding the right balance in dropout probability, or are there more complex interactions at play?

Curriculum Methods and Initial Dropout Probability:

Section 4.3.2 investigates the effect of curriculum methods (step-wise linear, epoch-wise linear, exponential) and initial dropout probability in MacDrop.
The authors find that step-based curriculum methods generally outperform epoch-based methods, and that a rapid decline in dropout probability can diminish MacDrop's effectiveness.
It might be useful to explore why step-based methods are superior. Is it because they provide a more granular control over the dropout process? Additionally, the paper mentions that a rapid decline in dropout probability can diminish effectiveness, but what is the optimal rate of decline, and how might this be determined?

Generalizability Across Architectures:

The paper evaluates MacDrop on several LLM architectures, but the sensitivity to massive weights appears to vary significantly across architectures (e.g., Gemma-2 is less sensitive).
While the authors acknowledge this, a deeper discussion on why certain architectures are more or less sensitive to massive weights would be valuable. Understanding the architectural features that influence this sensitivity could lead to more generalizable methods.

Impact on Different Tasks:

The paper shows that MacDrop improves performance on zero-shot downstream tasks and long context tasks, but has limited impact on generation tasks.
The authors provide some examples and discussion in Appendix G, but a more detailed analysis of why MacDrop is more effective in some tasks than others would be beneficial. Is it related to the type of knowledge or reasoning required by the task?

Computational Overhead:

The paper mentions that MacDrop introduces a negligible overhead (e.g., approximately 0.35 seconds per step for Llama-3-8B using LORA on 8xA100 GPUs).
While this sounds promising, it would be helpful to have a more detailed breakdown of where this overhead comes from and how it scales with larger models or different hardware.

**Methods And Evaluation Criteria:**

These methods and evaluation criteria appear to be appropriate for the problem and application:

- The proposed MacDrop method is designed to improve the fine-tuning of LLMs by reducing reliance on massive weights, and the curriculum dropout strategy makes sense in this context.

- The use of ablation studies to examine the effects of dropout scope, dropout probability scheduling, and curriculum methods is suitable for analyzing the impact of different components of MacDrop.

- The evaluation criteria cover various aspects of LLM performance, including perplexity, zero-shot accuracy on downstream tasks, long context understanding, and generation quality.

- The choice of benchmark datasets like WikiText, C4, PG-19, LongBench and Spec-Bench are standard in the field and provide a means to compare the effectiveness of the proposed method with existing approaches.

**Other Comments Or Suggestions:**

Typos and Grammatical Errors:
There are instances of typos and grammatical errors throughout the text. For example, in the abstract, "LLMS" should be "LLMs".
Attention to these details will improve the overall readability and professionalism of the paper.

Additional Analysis:
The paper could benefit from additional analysis in certain areas. For example, while the authors discuss the impact of MacDrop on zero-shot downstream tasks and long context tasks, a more detailed analysis of why MacDrop has limited impact on generation tasks would be valuable.

**Other Strengths And Weaknesses:**

Strengths:

- The paper provides a new perspective on understanding the internal mechanisms of LLMs by focusing on the role of massive weights in the intermediate state of feed-forward network modules. This approach is novel and contributes to the ongoing research on LLM interpretability.

- The paper is generally well-written and easy to follow. The authors provide clear explanations of their methodology, experimental setup, and results. The use of figures and tables effectively illustrates the key findings. The supplementary material provides additional details and analysis that further enhance the clarity and completeness of the paper.

Weaknesses:

- Limited Impact on Generation Tasks: The paper acknowledges that MacDrop has limited performance improvements in generation tasks. While the authors provide some analysis in Appendix G, a more in-depth investigation into why MacDrop is less effective for generation would be valuable.

- Generalizability Across Architectures: The paper demonstrates that the sensitivity to massive weights varies across different LLM architectures. While the authors discuss this, a deeper exploration of the architectural features that influence this sensitivity could lead to more generalizable methods.

**Questions For Authors:**

Details on Dropout Impact:

In Section 4.3.1, the paper mentions that strong dropout on massive weights deteriorates performance. Can the authors elaborate on why this happens and what the optimal balance is for dropout probability?

Generalizability of MacDrop:

MacDrop's effectiveness varies across different LLM architectures. Could the authors discuss the architectural features that make some LLMs more sensitive to massive weights than others?

**Relation To Broader Scientific Literature:**

In summary, the paper bridges the understanding of attention sinks and massive activations with an analysis of weight importance, and it leverages these insights to develop a more effective parameter-efficient fine-tuning method.

**Theoretical Claims:**

The paper does not include any proofs for theoretical claims. It focuses on empirical observations and proposing a practical method (MacDrop) based on those observations.

---

> ### Author Rebuttal · Authors · 2025-03-31
>
> Thank you for your in-depth review. We will update the manuscript to address the typos and grammatical errors, and suggested references.
>
> During the rebuttal process, we focus on addressing the questions in **Experimental Designs Or Analyses**, because this encompasses concerns raised in **Weaknesses**, **Comments Or Suggestions**, and **Questions**.
>
> ---
> ### Q1) Ablation Study On Dropout Scope (or, Details on Dropout Impact in **Questions**)
> - We conducted this ablation study to identify the proper dropout probability. Additionally, we emphasize that among three scopes, applying dropout to massive weights can only lead to improved performance.
> - We interpret the performance degradation under strong probabilities as follows:
>   - As demonstrated in the original dropout paper [1], excessively high dropout probabilities cause performance degradation due to **underfitting**. (Figure 9 in [1])
>   - Moreover, consider the extreme case where the dropout probability $p = 1.0$. This corresponds to training a model under the complete zeroing attack. Therefore, the gradients cannot capture meaningful update directions.
>
> ---
>
> ### Q2) Curriculum Methods and Initial Dropout Probability
> - As you mentioned, we consider training to be more stable when using the step-based curriculum, because it offers finer granularity compared to the epoch-based one, as shown in **Figure 7**. This is particularly relevant in our experimental setting (only 3 epochs fine-tuning). We believe that if training is conducted over sufficient epochs, the drawbacks of the epoch-based approach can be mitigated.
> - Next, while optimizing the curriculum is necessary, it is a highly challenging task and may incur significant additional overhead (Line 6 of Algorithm 1).
>   - Therefore, we proposed a general curriculum as a rule of thumb. Please refer to the last sentence of the last paragraph in **Section 4**. We believe that, at least in training environments similar to ours, the optimal curriculum likely lies between the Step and Exp. ($\alpha = 0.01$) strategies.
>
> ---
>
> ### Q3) Generalizability Across Architectures (or, Generalizability Across Architectures in **Weaknesses** and Generalizability of MacDrop in **Questions**)
> - We argue that the lower sensitivity of Phi-3.5-medium and Gemma-2 can be attributed to the dropout (in **Eq. 2**) and LayerNorm layers (in **Eq. 3**), respectively.
> - Regarding the Phi-3 family, we have observed that sensitivity varies with the model scale and the number of pre-training tokens. However, due to limitations in computational resources, we kindly ask for your understanding that we are currently unable to reproduce and verify this finding in detail.
> - However, for the Gemma-2 family, a very recent study [2] has provided an analysis indicating that the architectural design shown in **Eq. 3**—referred to as Peri-LN—preserves **gradient stability** by normalizing both input and output of each sub-layer and prevents the emergence of massive activations.
>   - We consider this paper to offer deeper insights into the functional role of LayerNorm, which we will cover in an updated manuscript.
>
> ---
> ### Q4) Impact on Different Tasks (or, Limited Impact on Generation Tasks in **Weaknesses** and Additional Analysis in **Comments Or Suggestions**)
>
> - Although we have considered this matter extensively, we have not been able to identify the exact cause.
> - As you have mentioned, we conducted an analysis of all tasks, including zero-shot, long context, and generation, at the sub-task level; however, the performance increase or decrease of MacDrop did not appear with any consistent characteristics.
> - Nevertheless, we believe that the lack of improvement of MacDrop in generation tasks may be due to their inherent complexity. We would greatly appreciate any suggestions or experimental designs that could help better isolate this phenomenon.
>
> ---
> ### Q5) Computational Overhead
> - Please refer to the last sentence of the last paragraph in **Section 3**. The comptational overhead comes from only masking and rollback processes in layer $l$, corresponding to the Lines 6-9 and 11-12 in Algorithm 1, respecitvely.
> - For Llama-3-70B, approximately 0.68 second per step is required. This overhead is related to the **dimension of the intermediate state** (e.g., 14,436 for 8B and 28,672 for 70B), because the number of massive indices is set to 5 for both the 8B and 70B models.
>   - Although it is nearly twice as much compared to 8B, the time required for loss computation (Line 10) has increased by approximately eight times, making addtional overhead even more negligible.
> - Unfortunately, we do not have different HW. However, it is expected that the overhead is neglectable compared to the loss calculation in other hardware as well.
>
> ---
>
> [1] Dropout: A Simple Way to Prevent Neural Networks from Overfitting
> [2] Peri-LN: Revisiting Layer Normalization in the Transformer Architecture

---

### Decision · Program_Chairs · 2025-05-01

**Decision:**

Reject

**Comment:**

The paper studies and identifies the phenomenon of "massive weights" in large language models which occur in the intermediate state of a feed-forward network module and contribute to the known massive activations in these models. The authors claim that these weights are important for performance on downstream tasks, and propose a method called MacDrop (massive weights curriculum dropout) to reduce the reliance on these massive weights during parameter-efficient fine-tuning.

Review summary: The paper received borderline reviews from all four reviewers. The reviewers noted that the identification of the massive weights phenomenon and the experiment demonstrating the top-k zeroing attack and its impact on downstream tasks is interesting. However, the proposed regularization, MacDrop, leads to mixed results. The overall contributions of the paper are weak.


Recommendation & justification: I recommend rejection as it does not meet the bar of acceptance to ICML.